# ΔFlow: An Efficient Multi-frame Scene Flow Estimation Method

**Qingwen Zhang**[1,✉]    **Xiaomeng Zhu**[1,3]    **Yushan Zhang**[2,✉]
**Yixi Cai**[1]    **Olov Andersson**[1]    **Patric Jensfelt**[1]

[1]KTH Royal Institute of Technology    [2]Linköping University    [3]Scania CV AB

qingwen@kth.se, yushan.zhang@liu.se

## Abstract

Previous dominant methods for scene flow estimation focus mainly on input from two consecutive frames, neglecting valuable information in the temporal domain. While recent trends shift towards multi-frame reasoning, they suffer from rapidly escalating computational costs as the number of frames grows. To leverage temporal information more efficiently, we propose DeltaFlow (ΔFlow), a lightweight 3D framework that captures motion cues via a Δ scheme, extracting temporal features with minimal computational cost, regardless of the number of frames. Additionally, scene flow estimation faces challenges such as imbalanced object class distributions and motion inconsistency. To tackle these issues, we introduce a Category-Balanced Loss to enhance learning across underrepresented classes and an Instance Consistency Loss to enforce coherent object motion, improving flow accuracy. Extensive evaluations on the Argoverse 2, Waymo and nuScenes datasets show that ΔFlow achieves state-of-the-art performance with up to 22% lower error and 2× faster inference compared to the next-best multi-frame supervised method, while also demonstrating a strong cross-domain generalization ability. The code is open-sourced at `https://github.com/Kin-Zhang/DeltaFlow` along with trained model weights.

## 1 Introduction

Scene flow estimation determines the 3D motion of each point between consecutive point clouds, making it an important task in computer vision and essential for autonomous driving [31, 3, 43, 41] and motion compensation [8, 46]. While early approaches focused on per-point feature learning [49, 47, 25, 26, 48, 40] and achieve high accuracy on small-scale datasets, they become computationally expensive when processing large-scale, high-density point clouds typical in autonomous driving. To reduce computational costs and enable real-time inference, recent methods [15, 44, 17, 28, 18] voxelize point features to estimate the scene flow vector field.

Meanwhile, real-world LiDAR data is acquired as a continuous stream rather than isolated frame pairs, making it crucial to leverage temporal information from multiple frames for more accurate motion estimation. To incorporate temporal information, existing multi-frame approaches either concatenate voxel features along the feature dimension [15, 44, 17] (see Figure 1(a)), or introduce an explicit temporal dimension to stack them [28, 18] (see Figure 1(b)). Both strategies lead to increased feature size and network parameters as the number of frames increases, resulting in higher memory consumption and slower training and inference.

To address these limitations, we propose DeltaFlow (ΔFlow), a computationally efficient 3D framework for multi-frame scene flow estimation. It applies a direct Δ scheme between voxelized frames, as shown in Figure 1(c), avoiding the feature concatenation or 4D stacking used by prior methods in

---

✉ Corresponding authors.

39th Conference on Neural Information Processing Systems (NeurIPS 2025).

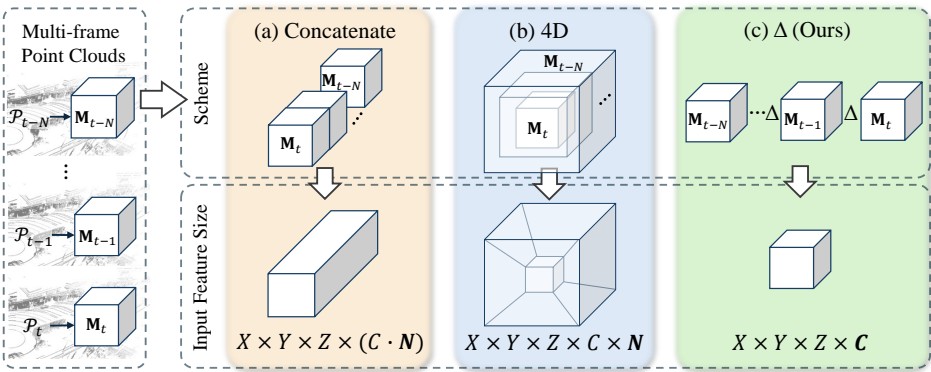

Figure 1: Comparison of multi-frame strategies for scene flow estimation. For clarity, voxelized features are shown in dense formats. $X, Y, Z$ denote spatial resolution, $C$ represents feature channels, and $N$ is the number of frames. Existing methods process voxelized representations using (a) Concatenation features along the channel dimension [44, 15]; (b) 4D methods stack features in an additional temporal dimension [18, 28]. Both increase input size as $N$ grows. (c) Our proposed $\Delta$Flow applies a $\Delta$ scheme between voxelized frame, maintaining a compact feature representation and a constant feature size independent of $N$.

Figure 1(a), (b). The scheme allows the network to focus on "*what is changing*" in the scene rather than the static background, aligning with the core objective of scene flow estimation. It also keeps the input feature size of $\Delta$Flow constant regardless of the number of frames, effectively addressing the scalability challenge in multi-frame scene flow estimation.

We further propose improvements to the scene flow supervision signal. Existing approaches [44, 15, 34] for driving scenarios primarily focus on distinguishing between static and moving objects. However, they do not explicitly address severe class imbalances (e.g., cars versus pedestrians) as mentioned in [16], nor ensure motion consistency across all points within the same object. To tackle these issues, we propose a Category-Balanced Loss to achieve more balanced training across all the classes, including the underrepresented classes (e.g., pedestrians, cyclists), and an Instance Consistency Loss to enforce a uniform motion for each individual instance.

$\Delta$Flow achieves the best performance for real-time scene flow estimation on Argoverse 2, Waymo and nuScenes datasets, outperforming the next-best multi-frame supervised method by up to 22%. It also demonstrates high computational efficiency and scalability in multi-frame settings, achieving up to $2\times$ faster inference, and exhibits strong generalization ability across different datasets. With its accuracy and efficiency, $\Delta$Flow is well-suited for real-world autonomous driving applications. The contributions of this paper are as follows: (1) We propose $\Delta$Flow, a lightweight 3D framework for multi-frame scene flow estimation that efficiently extracts motion cues by a $\Delta$ scheme between voxelized frames, maintaining a compact feature representation that is scalable in the temporal domain. (2) We introduce a Category-Balanced Loss and an Instance Consistency Loss to enhance dynamic flow for underrepresented classes and improve motion consistency for individual instances. (3) We demonstrate that $\Delta$Flow achieves state-of-the-art real-time performance on three datasets, while exhibiting high computational efficiency and strong cross-domain generalization ability.

## 2 Related Work

**Scene Flow Estimation** Scene flow estimation describes the 3D motion field between temporally successive point clouds [36, 23, 41, 14, 49]. Early methods focused on point-wise feature learning [38, 20, 37, 25, 47, 49], achieving high accuracy on small-scale synthetic datasets such as ShapeNet [6] and FlyingThings3D [29]. However, when applied to large-scale, high-density point clouds typical in autonomous driving [11, 33, 30, 5, 2, 10], these methods require downsampling due to high memory costs and are not well optimized for sequential data, limiting their practical utility.

To handle large-scale point clouds, FastFlow3D [15] voxelized point clouds and concatenated voxelized features from two frames before feeding them into the network. However, voxelization sacrifices fine-grained motion details, as point-to-voxel transformations reduce spatial resolution, thereby diminishing accuracy at the object level. DeFlow [44] addressed this by introducing GRU-

based voxel-to-point refinement, while SSF [17] leveraged sparse convolutions and virtual voxels to enhance long-range scene flow estimation.

**Multi-frame challenges**  Multi-frame modeling has become a key trend in scene flow estimation, as leveraging past frames provides richer temporal context and allows for a more comprehensive understanding of motion dynamics over time [18, 13, 35]. One approach is to process all frames in a sequence offline, as demonstrated by EulerFlow [35]. Although it achieves high accuracy, it is highly computationally demanding, requiring 24 hours to process a sequence of 157 frames in Argoverse 2, making it infeasible for real-time applications. An alternative is to concatenate multi-frame features, as done by most voxelized methods mentioned earlier. However, this leads to feature expansion, higher memory consumption, and limited temporal consistency as more frames are added.

To improve efficiency, two common strategies are used: 1) spatial optimization and 2) temporal optimization. For spatial optimization, methods in [15, 44, 17] reduce spatial dimensions by compressing the Z-dimension into a bird's-eye view (BEV) representation, effectively transforming the network into 2D processing. While this reduces the computational cost, it removes height information, which may degrade accuracy. Alternatively, Kim *et al.* [18] applies sparse voxelization, using storage formats like coordinate format (COO) to store only nonzero elements with an indices matrix for coordinates and a value array for features. This efficiently reduces memory consumption while preserving the full 3D structure. For temporal optimization, Kim *et al.* [18] introduces an explicit temporal dimension, avoiding direct feature concatenation across frames. Instead of increasing feature channels, it proposes a 4D network with separate 3D spatial and 1D temporal convolutions, scaling input size multiplicatively with the number of frames. This reduces feature expansion and enhances the feasibility of multi-frame processing.

In this work, we also enhance spatial efficiency with sparse voxelization, preserving 3D structure while reducing memory usage. For temporal efficiency, we introduce a $\Delta$ scheme that extracts motion cues without expanding feature size, addressing scalability challenges in multi-frame scene flow estimation.

**Other challenges**  Beyond computational challenges, scene flow estimation is further complicated by label imbalance. Most LiDAR points in autonomous driving scenarios belong to static structures such as buildings or roads, while dynamic objects are comparatively scarce. This imbalance biases model learning, making motion variations harder to capture. Prior works [44, 15, 34, 45] attempt to mitigate this issue using scaling functions in loss design to balance motion contributions. Among them, the motion-aware loss from DeFlow [44], which applies unweighted three-speed ranges, has shown the most effectiveness.

However, category imbalance remains a challenge. The recent scene flow evaluation metric [16] highlights that small but safety-critical categories, such as pedestrians and cyclists, are underrepresented compared to larger vehicle classes, making small-instance predictions challenging. Meanwhile, Zhang *et al.* [45] emphasize the need for object-level motion consistency, where instances within the same object should share coherent scene flow. To address these issues, we introduce a Category-Balanced Loss and Instance Consistency Loss in this paper.

## 3  Method

### 3.1  Problem Formulation

Given two consecutive point clouds, $\mathcal{P}_{t-1} \in \mathbb{R}^{N_{t-1} \times 3}$ and $\mathcal{P}_t \in \mathbb{R}^{N_t \times 3}$, scene flow estimation aims to predict how the points $p_{t-1} \in \mathcal{P}_{t-1}$ move from time $t-1$ to $t$, resulting in a scene flow vector field $\mathcal{F}_{t-1} \in \mathbb{R}^{N_{t-1} \times 3}$. The estimated flow $\hat{\mathcal{F}}_{t-1}$ from $\mathcal{P}_{t-1}$ to $\mathcal{P}_t$ can be decomposed as:

$$\hat{\mathcal{F}} = \mathcal{F}_{ego} + \Delta\hat{\mathcal{F}}, \tag{1}$$

where $\mathcal{F}_{ego}$ represents the motion caused by the ego vehicle. This motion is computed from the relative transformation of the sensor pose $\mathbf{T}_{ego}^{t-1 \to t}$ between time $t-1$ and $t$, i.e., $\mathcal{F}_{ego} = \mathbf{T}_{ego}^{t-1 \to t}\mathcal{P}_{t-1} - \mathcal{P}_{t-1}$. Since $\mathcal{F}_{ego}$ can be determined directly from odometry, the goal is to estimate the residual scene flow $\Delta\hat{\mathcal{F}}$ with our approach.

Most existing scene flow estimation methods focus on reasoning with two consecutive point cloud frames [21, 22, 44, 15, 17], learning a mapping: $\{\mathbf{T}_{ego}^{t-1 \to t}\mathcal{P}_{t-1}, \mathcal{P}_t\} \to \Delta\mathcal{F}_{t-1}$. With the recent trend shifting towards multi-frame reasoning [18, 46, 35], we instead focus on learning: $\{\mathbf{T}_{ego}^{t-N \to t}\mathcal{P}_{t-N}, ..., \mathbf{T}_{ego}^{t-1 \to t}\mathcal{P}_{t-1}, \mathcal{P}_t\} \to \Delta\mathcal{F}_{t-1}$ to leverage additional past frames for improved motion estimation.

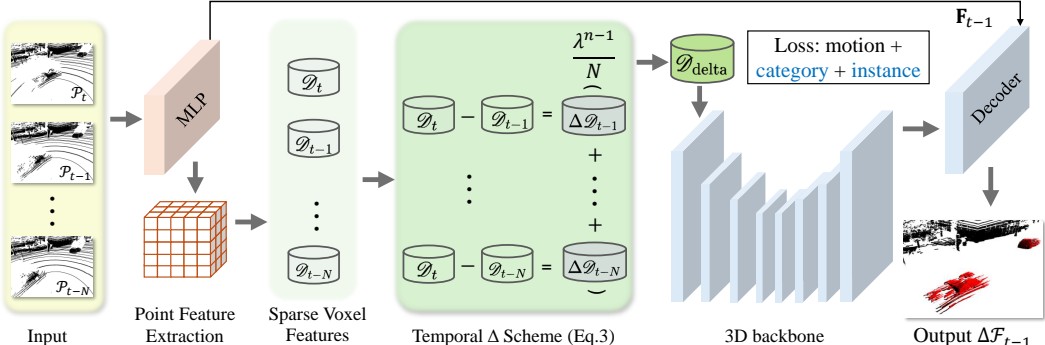

Figure 2: Overview of the proposed $\Delta$Flow architecture. The framework first extracts point-level features and voxelize them to obtain sparse voxel features $\mathscr{D}$. The core temporal $\Delta$ scheme then computes the difference between the current frame $t$ and previous frames $(t-1, \ldots, t-N)$, weighted by a time-decay factor $\lambda$. The resulting $\Delta$ feature $\mathscr{D}_{\text{delta}}$ is then passed to a 3D backbone–decoder network to estimate the final scene flow $\Delta\mathcal{F}_{t-1}$. This approach captures motion-specific cues efficiently while keeping the architecture compact and scalable, regardless of the number of frames.

## 3.2 $\Delta$Flow

To effectively learn multi-frame information, we propose $\Delta$Flow, as shown in Figure 2. We extract a sparse $\Delta$ feature that efficiently encodes temporal and spatial information from the frames, then feed it into a standard backbone-decoder network for scene flow estimation.

**Point Feature Extraction**  Following common practices in scene flow estimation [15, 44, 18], we first encode individual point clouds using PointPillars [19], generating point-wise feature representations $\{\mathbf{F}_{t-N}, ..., \mathbf{F}_{t-1}, \mathbf{F}_t\}$. Each $\mathbf{F}_t \in \mathbb{R}^{N_t \times C}$ represents the feature embedding for $\mathcal{P}_t$, where $N_t$ is the number of points and $C$ is the feature dimension.

**Sparse Spatial Representation**  We then encode the point features into a sparse 3D representation, which processes only non-empty voxels to reduce computational overhead in large-scale 3D grids. Given encoded point features $\mathbf{F} \in \mathbb{R}^{N_{\mathcal{P}} \times C}$, we construct sparse voxel features $\mathscr{D} \in \mathbb{R}^{V \times C}$ through point-to-voxel aggregation:

$$\mathscr{D}[v_i] = \begin{cases} \dfrac{\sum_{p \in \mathcal{P}^{v_i}} \mathbf{f}_p}{|\mathcal{P}^{v_i}|} & v_i \in \mathcal{V}, \\ \mathbf{0} & \text{otherwise}, \end{cases} \tag{2}$$

where $v_i = (x_i, y_i, z_i)$ denotes the $i$-th active voxel coordinate in $\mathcal{V}$ (the set of non-empty voxels with $|\mathcal{V}| = V$). $\mathcal{P}^{v_i}$ represents points inside voxel $v_i$ and $\mathbf{f}_p \in \mathbb{R}^C$ is the feature of point $p$.

**Temporal $\Delta$ Scheme**  In order to extract motion signals from the sparse voxel features, we propose a simple yet effective $\Delta$ scheme comprising subtraction, temporal weighting and summation steps:

$$\mathscr{D}_{\text{delta}} = \sum_{n=1}^{N} \lambda^{n-1} (\mathscr{D}_t - \mathscr{D}_{t-n})/N, \tag{3}$$

resulting in the $\Delta$ feature $\mathscr{D}_{\text{delta}} \in \mathbb{R}^{V \times C}$. $N$ is the number of past frames considered, and $\lambda$ applies a temporal decay to earlier frames.

The $\Delta$ scheme is designed to extract global motion patterns from multiple frames while maintaining computational efficiency. First, voxel-wise differences between the current frame and previous ones are computed, encouraging the model to focus on what is changing in the scene while minimizing reliance on static features. These differences are then weighted by a decay factor $\lambda \in (0, 1]$, which assigns higher importance to more recent frames. The weighted differences are subsequently summed to accumulate the trail of moving objects and produce a temporal representation that captures long-term motion information. The intuition for this accumulation is to mimic how humans can interpret motion in a single image from motion blur. We provide qualitative support for the design of the $\Delta$ scheme in Section 5.5, which illustrates the temporal behavior of the proposed method and highlights its emphasis on motion-related changes.

Notably, this scheme maintains a constant feature dimension, ensuring that the feature size remains unaffected by the number of frames. This allows the model to process an arbitrary number of past frames without increasing computational overhead in the backbone-decoder network.

**Backbone-Decoder network**    The $\Delta$ feature $\mathscr{D}_{\text{delta}}$ is then served as input to a 3D backbone-decoder network for scene flow estimation. It is first processed by a 3D backbone for feature extraction:

$$\mathscr{D}_{(\text{out})} \ = \ \text{Backbone}\Big( \mathscr{D}_{\text{delta}}; \mathbf{W}_{\text{net}} \Big), \tag{4}$$

where Backbone refers to any network capable of processing sparse 3D voxel inputs, and $\mathbf{W}_{\text{net}}$ are trainable network parameters. The backbone output $\mathscr{D}_{(\text{out})}$ is then sent to a scene flow decoder network to estimate a per-point 3D scene flow vector:

$$\Delta\hat{\mathcal{F}} = \text{Decoder}\big( \text{V2P}(\mathscr{D}_{(\text{out})}), \mathbf{F}_{t-1}; \mathbf{W}_d \big),$$

where the mapping $\text{V2P}(\mathscr{D}_{(\text{out})}) : \text{R}^{V \times C} \to \text{R}^{N_{t-1} \times C}$ maps features back to points via pre-recorded coordinate indexing, and $\mathbf{W}_d$ are trainable decoder weights.

## 3.3    Loss Function

We employ three loss functions to supervise scene flow estimation: the motion-awareness loss from DeFlow [44], and two new ones proposed in this work, a category-balanced loss and an instance consistency loss, that address class imbalance and motion inconsistency.

**Motion-awareness Loss**    The motion-awareness loss $\mathcal{L}_{\text{deflow}}$ from DeFlow [44] is designed to mitigate data imbalance between static and dynamic points by prioritizing dynamic point flow estimation. It categorizes points in $\mathcal{P}_t$ based on their motion speed into three groups[1]: $\{\mathcal{P}_{t/1}, \mathcal{P}_{t/2}, \mathcal{P}_{t/3}\}$, resulting in the loss:

$$\mathcal{L}_{\text{deflow}} = \sum_{i=1}^{3} \frac{1}{|\mathcal{P}_{t/i}|} \sum_{p \in \mathcal{P}_{t,s}} \left\| \Delta\hat{\mathcal{F}}(p) - \Delta\mathcal{F}_{\text{gt}}(p) \right\|_2. \tag{5}$$

**Category-Balanced Loss**    The Category-Balanced Loss is designed to categorize dynamic objects based on class and motion speed, ensuring a more balanced scene flow learning across different object categories. To achieve this, we assign to each point $p$ a meta-category from the set $\mathcal{C}$, with each category $c$ assigned a predefined weight $w_c$. The loss is then defined as:

$$\mathcal{L}_C = \sum_{c \in \mathcal{C}} w_c \sum_{b \in \mathcal{B}} \gamma_b \frac{1}{|\mathcal{P}_{c,b}|} \sum_{p \in \mathcal{P}_{c,b}} \left\| \Delta\hat{\mathcal{F}}(p) - \Delta\mathcal{F}_{\text{gt}}(p) \right\|_2, \tag{6}$$

where $\gamma_b$ are speed-dependent coefficients that adjust the weighting based on motion speed.

**Instance Consistency Loss**    The instance consistency loss is designed to ensure that all points on a rigid object exhibit a consistent scene flow. To achieve this, we let $\mathcal{I}$ be the set of object instances, and for each instance $I \in \mathcal{I}$, let $\mathcal{P}_I$ denote the set of points belonging to $I$. The per-instance average estimated error is defined as:

$$\hat{e}_I = \sum_{p \in \mathcal{P}_I} \frac{||\Delta\hat{\mathcal{F}}(p) - \Delta\mathcal{F}_{\text{gt}}(p)||_2}{|\mathcal{P}_I|}.$$

Each instance is assigned a representative meta-category $c_I$, and only moving instances ($\mathcal{I}'$) where speed exceeds $0.4 \, \text{m/s}$ are considered. The loss is then defined as:

$$\mathcal{L}_I = \frac{1}{|\mathcal{I}'|} \sum_{I \in \mathcal{I}'} \omega_{c_I} \hat{e}_I \exp\left(\hat{e}_I\right). \tag{7}$$

The final loss function is the sum of all three losses:

$$\mathcal{L}_{\text{total}} = \mathcal{L}_{\text{deflow}} + \mathcal{L}_C + \mathcal{L}_I. \tag{8}$$

---

[1]DeFlow loss categorizes motion speed into three groups $\mathcal{B}$: 0 to 0.4 m/s, 0.4 to 1.0 m/s, and above 1.0 m/s, respectively.

# 4 Experiments Setup

## 4.1 Datasets

Experiments are conducted on three commonly used large-scale autonomous driving datasets in scene flow estimation: Argoverse 2 [39], which employs two roof-mounted 32-channel LiDARs; Waymo [33], which uses a single 64-channel LiDAR; and nuScenes [5], which uses a 32-channel LiDAR. Ground removal is applied to Argoverse 2 and Waymo using HDMap information following [34], and to nuScenes using line-fit ground segmentation [12].

**Argoverse 2** provides an official public scene flow challenge [1], consisting of 700 training and 150 validation scenes, each lasting 15 seconds at 10 Hz, totaling 110,071 point cloud frames. An additional 150 test scenes are available for evaluation via the online leaderboard. The scene flow ground truth is generated from human annotations with tracking-based inference to estimate 3D motion. However, as noted in [46], non-ego motion distortion can lead to inaccurate annotations, as points from fast-moving objects may fall outside their labeled bounding boxes. To address this issue, we follow the velocity-aware annotation refinement strategy in [46], which enlarges the bounding box of each object along its motion direction to include all distorted points. As illustrated in Figure 3, the 3D flow vectors (red lines) accurately capture the true motion of previously distorted points.

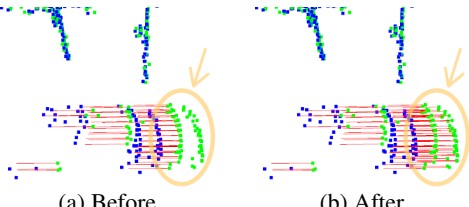

(a) Before      (b) After

Figure 3: Comparison of scene flow ground truth before and after motion compensation on a high-speed car. Blue points: LiDAR scan at $t+1$; Green points: LiDAR scan at $t$; Red lines: annotated flow vectors.

**Waymo** [15, 33] contains 798 training and 202 validation sequences, each recorded at 10 Hz for around 20 seconds. The training set consists of 155,000 frames. Motion distortion affects both the point cloud and ground truth. However, direct annotation refinement is infeasible as it does not provide per-point timestamps. Thus, evaluation is conducted using the original annotations.

**nuScenes** [5] includes 700 training and 150 validation scenes, each recorded at 20 Hz for around 20 seconds. The training set contains 275,150 frames, of which 27,392 ($\approx$10%) are annotated with ground-truth labels, yielding an effective annotation rate of 2 Hz. For consistency with Argoverse 2 and Waymo, we downsample the LiDAR data to 10 Hz, yielding a standard 100 ms interval between consecutive frames. Ground-truth scene flow is constructed by computing a rigid transformation for each object from its annotated 3D bounding box and velocity, then applying this transformation to all points within the object to obtain flow vectors.

## 4.2 Evaluation Metrics

The leaderboard [1] evaluates scene flow using two metrics: three-way End Point Error (EPE) and Dynamic Bucket-Normalized EPE. EPE is defined as the L2 norm of the difference between predicted and ground truth flow vectors, measured in centimeters. **Three-way EPE** [7] computes the unweighted average EPE over three regions: foreground dynamic (FD), foreground static (FS), and background static (BS). A point is classified as dynamic if its ground truth velocity exceeds 0.5 m/s, and foreground if it lies within the bounding box of any tracked object. **Dynamic Bucket-Normalized EPE** [16] groups point into predefined motion buckets based on their speeds and normalizes EPE by mean speed ($\frac{\text{Mean EPE}}{\text{Mean speed}}$). This metric evaluates four object categories: regular cars (CAR), other vehicles (OTHER) such as trucks and buses, pedestrians (PED), and wheeled vulnerable road users (VRU), including bicycles and motorcycles.

## 4.3 Implementation Details

In our implementation, we adopt MinkowskiNet [9] as our 3D backbone, a widely used sparse convolutional network known for strong performance in 3D perception tasks. The scene flow decoder follows DeFlow [44], enabling effective per-point scene flow prediction. Further details on the $\Delta$ scheme and the full 3D backbone-decoder architecture are provided in Appendix A.1.

For Argoverse 2, test set results are obtained directly from the public leaderboard [1] to ensure a fair comparison. For Waymo and other local experiments, all baselines are retrained and reproduced with ego-motion compensation under identical device settings for consistent evaluation. Training

Table 1: Performance comparisons on Argoverse 2 test set from the public leaderboard [1]. Upper groups are self-supervised methods, lower are supervised methods. Our method achieves state-of-the-art performance in scene flow estimation. '#F' denotes the number of input frames. Runtime is reported per sequence (around 157 frames), with '-' indicating unreported runtime. 's', 'm', and 'h' represent seconds, minutes, and hours, respectively. Purple highlighted runtimes indicate offline methods. **Bold** and underline mark the best and second-best results.

| Methods | #F | Runtime per seq | Dynamic Bucket-Normalized ↓ | | | | | Three-way EPE (cm) ↓ | | | |
|---|---|---|---|---|---|---|---|---|---|---|---|
| | | | Mean | CAR | OTHER | PED | VRU | Mean | FD | FS | BS |
| Ego Motion Flow | - | - | 1.000 | 1.000 | 1.000 | 1.000 | 1.000 | 18.13 | 53.35 | 1.03 | 0.00 |
| SeFlow [45] | 2 | 7.2s | 0.309 | 0.214 | 0.291 | 0.464 | 0.265 | 4.86 | 12.14 | 1.84 | 0.60 |
| ICP Flow [24] | 2 | - | 0.331 | 0.195 | 0.331 | 0.435 | 0.363 | 6.50 | 13.69 | 3.32 | 2.50 |
| ZeroFlow [34] | 3 | 5.4s | 0.439 | 0.238 | 0.258 | 0.808 | 0.452 | 4.94 | 11.77 | 1.74 | 1.31 |
| FastNSF [22] | 2 | 12m | 0.383 | 0.296 | 0.413 | 0.500 | 0.322 | 11.18 | 16.34 | 8.14 | 9.07 |
| NSFP [21] | 2 | 1.0h | 0.422 | 0.251 | 0.331 | 0.722 | 0.383 | 6.06 | 11.58 | 3.16 | 3.44 |
| Floxels [13] | 13 | 24m | 0.154 | 0.112 | 0.213 | 0.195 | 0.097 | 4.73 | 10.30 | 3.65 | **0.24** |
| EulerFlow [35] | all | 24h | 0.130 | 0.093 | 0.141 | 0.195 | **0.093** | 4.23 | 4.98 | 2.45 | 5.25 |
| FastFlow3D [15] | 2 | 5.4s | 0.532 | 0.243 | 0.391 | 0.982 | 0.514 | 6.20 | 15.64 | 2.45 | 0.49 |
| TrackFlow [16] | - | - | 0.269 | 0.182 | 0.305 | 0.358 | 0.230 | 4.73 | 10.30 | 3.65 | **0.24** |
| DeFlow [44] | 2 | 7.2s | 0.276 | 0.113 | 0.228 | 0.496 | 0.266 | 3.43 | 7.32 | 2.51 | 0.46 |
| SSF [17] | 2 | 5.2s | 0.181 | 0.099 | 0.162 | 0.292 | 0.169 | 2.73 | 5.72 | 1.76 | 0.72 |
| Flow4D [18] | 2 | 12.8s | 0.174 | 0.095 | 0.167 | 0.278 | 0.155 | 2.51 | 5.73 | 1.48 | 0.30 |
| | 5 | 15s | 0.145 | 0.087 | 0.150 | 0.216 | 0.127 | 2.24 | 4.94 | **1.31** | 0.47 |
| ΔFlow (Ours) | 2 | 7.6s | 0.145 | 0.084 | 0.144 | 0.225 | 0.125 | 2.30 | 4.81 | 1.44 | 0.66 |
| | 5 | 8s | **0.113** | **0.077** | **0.129** | **0.149** | 0.096 | **2.11** | **4.33** | 1.37 | 0.64 |

Table 2: Comparisons on Waymo validation set where each sequence contains around 200 frames. Upper groups are self-supervised methods, lower are supervised methods.

| Methods | Runtim per seq | Three-way EPE (cm) ↓ | | | |
|---|---|---|---|---|---|
| | | Mean | FD | FS | BS |
| SeFlow [45] | 14.8s | 5.98 | 15.06 | 1.81 | 1.06 |
| ZeroFlow [34] | 12.4s | 8.52 | 21.62 | 1.53 | 2.41 |
| NSFP [21] | 1.6h | 10.05 | 17.12 | 10.81 | 2.21 |
| FastFlow3D [15] | 12.4s | 7.84 | 19.54 | 2.46 | 1.52 |
| DeFlow [44] | 14.8s | 4.46 | 9.80 | 2.59 | 0.98 |
| Flow4D [18] | 33s | 2.03 | 4.82 | 0.78 | **0.49** |
| ΔFlow (Ours) | 18s | **1.64** | **4.04** | **0.29** | 0.58 |

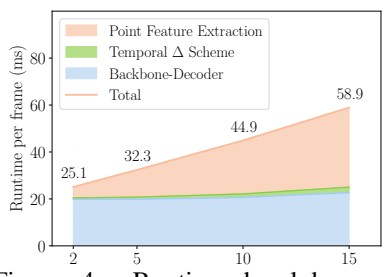

Figure 4: Runtime breakdown of ΔFlow on Argoverse 2 validation set as the number of input frames varies from 2 to 15 (x-axis).

resources and settings are detailed in Appendix A.2. The code is open-sourced at `https://github.com/Kin-Zhang/DeltaFlow` along with trained model weights.

## 5 Results and Discussion

### 5.1 State-of-the-art Comparison

The Argoverse leaderboard test set results are summarized in Table 1. The proposed ΔFlow achieves the best mean EPE and dynamic bucket-normalized EPE scores among all methods, demonstrating both high accuracy and efficiency. It reduces mean dynamic bucket-normalized EPE by 13% compared to the second-best method, EulerFlow, and by 22% compared to Flow4D, while running twice as fast as Flow4D and thousands of times faster than EulerFlow. Even with two frames, ΔFlow outperforms all methods using the same setting, indicating that the Δ scheme effectively encodes motion differences between frames without subtracting out the important information. This state-of-the-art performance is consistently validated on other large-scale datasets. On the Waymo dataset (shown in Table 2), ΔFlow again achieves the lowest mean EPE, outperforming the next-best model Flow4D by 19% while being 45% faster. On the nuScenes validation set (shown in Table 3), ΔFlow establishes a new benchmark by a significant margin, reducing the mean EPE by 39% compared to Flow4D. This consistently leading performance across datasets with different LiDAR configurations highlights the robustness and generalization capability of our approach.

Table 3: Comparisons on nuScenes validation set with a 10Hz LiDAR frequency, where each sequence contains around 200 frames. Upper groups are self-supervised methods, lower are supervised methods.

| Methods | #F | Runtime per seq | Dynamic Bucket-Normalized ↓ | | | | | Three-way EPE (cm) ↓ | | | |
|---|---|---|---|---|---|---|---|---|---|---|---|
| | | | Mean | CAR | OTHER | PED | VRU | Mean | FD | FS | BS |
| Ego Motion Flow | - | - | 1.000 | 1.000 | 1.000 | 1.000 | 1.000 | 12.34 | 35.94 | 1.07 | 0.00 |
| SeFlow [45] | 2 | 6s | 0.544 | 0.396 | 0.635 | 0.726 | 0.419 | 8.19 | 16.15 | 3.97 | 4.45 |
| FastNSF [22] | 2 | 2.6m | 0.560 | 0.436 | 0.523 | 0.737 | 0.543 | 12.16 | 18.20 | 6.11 | 12.18 |
| NSFP [21] | 2 | 3.5m | 0.602 | 0.463 | 0.456 | 0.829 | 0.662 | 10.79 | 20.26 | 4.88 | 7.23 |
| DeFlow [44] | 2 | 6s | 0.314 | 0.163 | 0.286 | 0.533 | 0.275 | 3.98 | 6.99 | 3.45 | 1.50 |
| Flow4D [18] | 5 | 9s | 0.279 | 0.204 | 0.312 | 0.379 | 0.222 | 3.82 | 8.05 | 1.82 | 1.58 |
| ΔFlow (Ours) | 5 | 7s | **0.216** | **0.138** | **0.219** | **0.327** | **0.181** | **2.33** | **4.83** | **1.37** | **0.79** |

Table 4: Scalability comparison of multi-frame scene flow estimation on the Argoverse 2 validation set. '#F' denotes the number of input frames processed. Flow4D and our ΔFlow are evaluated across increasing frame counts, reporting relative training speed, memory usage, and bucket-normalized accuracy.

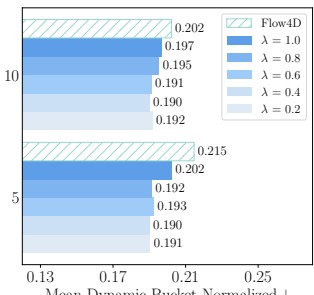

Figure 5: Ablation study of the decay factor $\lambda$ in ΔFlow, evaluated with 5 and 10 input frame settings.

| Method | #F | Speed ↑ | Memory ↓ | Dynamic Bucket-Normalized ↓ | | | | |
|---|---|---|---|---|---|---|---|---|
| | | | | Mean | CAR | OTHER | PED | VRU |
| Flow4D | 2 | 1.03× | 1.20× | 0.2269 | 0.1648 | 0.1738 | 0.2960 | 0.2729 |
| | 5 | 0.65× | 1.85× | 0.2147 | 0.1631 | 0.1767 | 0.2522 | 0.2667 |
| | 10 | 0.37× | 2.82× | 0.2022 | 0.1494 | **0.1707** | 0.2284 | 0.2603 |
| | 15 | 0.26× | 3.7× | 0.2055 | 0.1593 | 0.1738 | 0.2280 | 0.2607 |
| ΔFlow | 2 | **1.04×** | **0.98×** | 0.2116 | 0.1543 | 0.1751 | 0.2723 | 0.2449 |
| | 5 | 1.00× | 1.00× | **0.1901** | **0.1479** | 0.1723 | 0.2160 | 0.2243 |
| | 10 | 0.68× | 1.2× | **0.1901** | 0.1500 | 0.1853 | 0.2010 | **0.2241** |
| | 15 | 0.51× | 1.22× | 0.1916 | 0.1511 | 0.1911 | **0.2001** | 0.2242 |

## 5.2 Multi-frame Analysis

To further evaluate ΔFlow in multi-frame settings, we analyze its efficiency, scalability, and performance as frames increase, comparing it to Flow4D [18]. We also assess the effect of the time decay factor $\lambda$ in the $\Delta$ scheme on multi-frame processing.

**Efficiency and scalability** Table 4 compares the computational cost and accuracy of ΔFlow and Flow4D across different frame counts. As the number of input frames increases, Flow4D experiences a sharp drop in speed and a significant rise in memory consumption. By 15 frames, it requires 3.7× more memory and runs at only 0.26× speed, making large-frame modeling impractical. In contrast, ΔFlow scales more efficiently, with only 1.22× memory growth and twice the speed of Flow4D at higher frame counts. Notably, the slowdown of ΔFlow mainly stems from the point feature extraction for each additional frame, as shown in the runtime breakdown in Figure 4, while the temporal $\Delta$ scheme and backbone-decoder network add negligible cost. As a result, ΔFlow enables multi-frame processing without excessive computational overhead. This efficiency also validates the scalability of the $\Delta$ scheme, where the $\Delta$ feature maintains a constant feature size across frames, preventing the feature expansion problem seen in prior methods. Additional analysis on $\Delta$ scheme efficiency is provided in Appendix B.1.

**Performance** Beyond computational efficiency, both ΔFlow and Flow4D achieve lower mean dynamic bucket-normalized EPE with 5 or 10 frames compared to only 2 frames in Table 4, indicating that multi-frame modeling improves performance. However, this improvement diminishes or even declines at 15 frames. A likely reason is that long-ago frames become less informative for predicting current motion, and incorporating such outdated context may introduce noise. This highlights an open challenge in optimizing long-term temporal information usage in real-time to maximize scene flow performance.

**Time decay factor** We evaluate different decay factor $\lambda$ values (in Eq. (3)) for 5-frame and 10-frame settings and summarize the results in Figure 5 to analyze their impact on multi-frame processing. Overall, applying a decay $\lambda < 1$ consistently improves mean dynamic bucket-normalized performance over the non-decayed $\lambda = 1$ baseline, as it emphasizes recent frames and downweighting older ones.

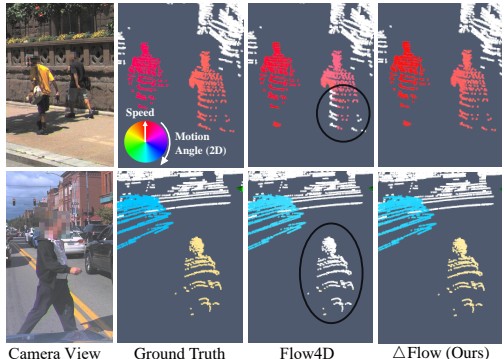

Figure 6: Qualitative comparison on the Argoverse 2. The left column displays camera views for reference, while the right columns visualize scene flow predictions, where Hue encodes direction and saturation represents magnitude. Our method, ΔFlow, produces more accurate and consistent flow estimates than the prior SOTA, Flow4D, particularly for small objects. (Best viewed in color.)

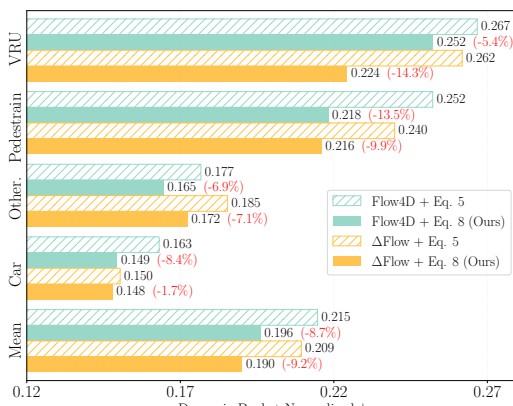

Figure 7: Ablation on our proposed loss (Eq. (8)) compared to the baseline (Eq. (5)) for both Flow4D and ΔFlow. Red percentages indicate relative error reduction compared to the baseline loss. Bucket-normalized error across dynamic categories shows consistent improvements, especially for smaller classes like pedestrians and vulnerable road users.

Notably, even without decay, ΔFlow surpasses Flow4D [18], demonstrating the robustness of our approach, with temporal decay providing additional performance gains. Additional per-category results for different $\lambda$ are provided in Appendix B.2.

### 5.3 Loss Analysis

To assess the impact of our proposed loss functions on ΔFlow, we analyze its performance across different object categories. As shown in Table 1, we observe that ΔFlow significantly improves pedestrian and wheeled object accuracy, reducing pedestrian error by 23.6% compared to the next-best competitor, EulerFlow, demonstrating its ability to better capture small dynamic instances. Additionally, qualitative comparisons in Figure 6 further show that ΔFlow produces more accurate motion vectors than Flow4D, particularly for small objects like pedestrians, with points of the same pedestrian exhibiting greater motion consistency. These improvements might come from the contribution of the Category-Balanced Loss and Instance Consistency Loss.

To validate this, we compare our proposed loss functions Eq. (8) with the baseline loss Eq. (5), evaluating both Flow4D and ΔFlow, as shown in Figure 7. The results demonstrate that our loss consistently improves bucket-normalized accuracy across all dynamic categories, particularly for underrepresented small objects. The improvements are observed across both models, with mean, pedestrian, and VRU errors reduced by approximately 10%, confirming the general effectiveness of our loss design. Furthermore, while improving dynamic bucketed-normalized performance, our loss preserves static scene information, as three-way mean EPE scores remain stable. More results and ablation studies on each loss item are provided in Appendix B.3.

Table 5: Cross-domain generalization measured by the three-way EPE (cm) metric. Each model is trained on one dataset (LiDAR number and channel in parentheses) and evaluated on another with a different LiDAR channel. Lower EPE indicates better generalization. Our proposed ΔFlow achieves the best performance in both evaluations.

| Methods | Three-way EPE (cm) ↓ | | | | | | | |
|---|---|---|---|---|---|---|---|---|
| | Argoverse 2 (2x32) → Waymo (64) | | | | Waymo (64) → Argoverse 2 (2x32) | | | |
| | Mean | FD | FS | BS | Mean | FD | FS | BS |
| SeFlow [45] | 5.98 | 15.06 | 1.81 | 1.06 | 6.29 | 15.56 | **1.16** | 2.16 |
| NSFP [21] | 10.05 | 17.12 | 10.81 | 2.21 | 6.81 | 13.28 | 3.43 | 3.71 |
| DeFlow [44] | 4.47 | 11.39 | 1.51 | 0.51 | 4.50 | 10.74 | 2.01 | 0.75 |
| Flow4D [18] | 3.33 | 8.31 | 0.92 | 0.75 | 4.01 | 9.56 | 1.74 | **0.74** |
| ΔFlow (Ours) | **3.12** | **7.91** | **0.77** | **0.67** | **3.24** | **7.12** | 1.57 | 1.02 |

## 5.4 Cross-domain Generalization

$\Delta$Flow also demonstrates strong cross-domain generalization, performing well when trained on one dataset (Argoverse 2 or Waymo) and tested on the other, as shown in Table 9. This setting is particularly challenging due to differences in sensor channels, point density, and scene distribution. To evaluate the cross-domain capability of $\Delta$Flow, we include self-supervised methods such as SeFlow and NSFP as baselines, which are trained or optimized directly on the unlabeled target domain. $\Delta$Flow achieves state-of-the-art performance compared to both the baselines and other supervised methods, with a three-way EPE of approximately 3cm and a foreground dynamic EPE of around 7cm. This strong performance is consistent across different object categories. A detailed breakdown of the dynamic bucket-normalized EPE per category is provided in Appendix B.4.

## 5.5 Visualization of the $\Delta$ Feature

To understand the effectiveness of our $\Delta$ scheme, we visualize feature maps from the $\Delta$ feature $\mathscr{D}_{\text{delta}}$, single-frame features ($\mathscr{D}_t$, $\mathscr{D}_{t-1}$, $\mathscr{D}_{t-2}$), and the final backbone output $\mathscr{D}_{(\text{out})}$, as shown in Figure 8. Each map is rendered by selecting the most activated channel after a max projection along the z-axis. Compared to single-frame features, $\mathscr{D}_{\text{delta}}$ focuses on "what is changing" in the scene. Zoom-in views (i) and (ii) in $\mathscr{D}_{\text{delta}}$ show trail-like activations on moving vehicles, confirming that the $\Delta$ feature effectively captures dynamic cues. View (iii) highlights static background noise, which is largely suppressed in $\mathscr{D}_{(\text{out})}$. This demonstrates that after passing through the backbone, motion cues are further refined while irrelevant static context is filtered out. These visualizations confirm that the $\Delta$ scheme guides the model to attend to dynamic regions, enabling cleaner and more robust motion representations.

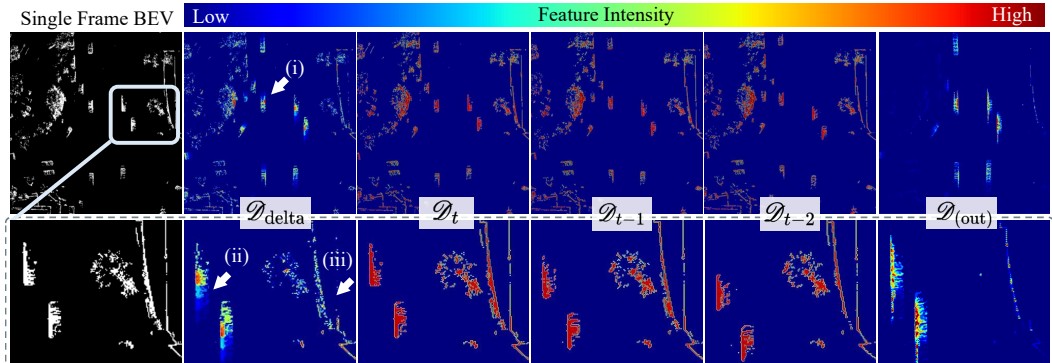

Figure 8: Visualization of feature maps. Top: BEV projections; bottom: zoomed-in views. The first column shows the BEV map of the point cloud $\mathcal{P}_t$, while the remaining columns visualize normalized feature maps from $\mathscr{D}_{\text{delta}}$, $\mathscr{D}_t$, $\mathscr{D}_{t-1}$, $\mathscr{D}_{t-2}$ and the final backbone output $\mathscr{D}_{(\text{out})}$. Each map is rendered by selecting the most activated channel after applying a max projection along the z-axis. Compared to the single-frame features, $\mathscr{D}_{\text{delta}}$ emphasizes regions with motion, such as moving vehicles in panels (i) and (ii), while downplaying static structures like buildings in panel (iii), highlighting its motion-centric design. The final feature, $\mathscr{D}_{(\text{out})}$, shows that the network further refines these cues.

## 6 Conclusion

This paper introduces $\Delta$Flow, an efficient framework for multi-frame scene flow estimation. It addresses the scalability challenge by leveraging a $\Delta$ scheme to extract motion cues without feature expansion as the number of frames grows. $\Delta$Flow achieves state-of-the-art performance on the Argoverse 2 and Waymo datasets while maintaining low computational cost and strong cross-domain generalization. Additionally, the proposed Category-Balanced Loss and Instance Consistency Loss enhance learning for underrepresented small objects and enforce coherent object-level motion. With its accuracy and efficiency, $\Delta$Flow is well-suited for real-world autonomous driving applications.

**Limitations and Future Work** While $\Delta$Flow offers a highly efficient multi-frame pipeline for scene flow estimation, it currently relies on ground-truth annotations for supervised training. A promising direction for future work is to incorporate self-supervised learning strategies into the $\Delta$Flow framework. This could enable high-efficiency, multi-frame scene flow estimation without labeled data, and enhance the real-time capabilities of self-supervised methods.

**Acknowledgement** We thank Chenhan Jiang and Yunqi Miao for helpful discussions during revision. This work was partially supported by the Wallenberg AI, Autonomous Systems and Software Program (WASP) funded by the Knut and Alice Wallenberg Foundation. This work was also in part financially supported by Digital Futures. The computations were enabled by the supercomputing resource Berzelius provided by National Supercomputer Centre at Linköping University and the Knut and Alice Wallenberg Foundation, Sweden.

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

# A Implementation Details

## A.1 Method

**Temporal $\Delta$ Scheme** The $\Delta$ scheme in the sparse implementation, shown in Algorithm 1, leverages CUDA memory coalescing and bank conflict avoidance techniques for optimal efficiency.

---

**Algorithm 1** $\Delta$ Scheme Implementation

---

    **Notation**: $\mathscr{D} = (\mathcal{V}, \mathbf{F})$ includes active voxel coordinate set $\mathcal{V}$ and corresponding feature vector $\mathbf{F}$.
1: **function** SPARSEDELTA($\mathscr{D}_A, \mathscr{D}_B, \mathtt{op}$)                           $\triangleright$ $\mathtt{op} \in \{\oplus, \ominus\}$
2:     $(\mathcal{V}_\cup, \mathbf{F}_\cup) \leftarrow \mathtt{sort\_by\_key}([\mathcal{V}_A, \mathcal{V}_B], [\mathbf{F}_A, \mathtt{op}\mathbf{F}_B])$
3:     $\mathcal{V}_\Delta, \mathbf{F}_\Delta \leftarrow \mathtt{reduce\_by\_key}(\mathcal{V}_\cup, \mathbf{F}_\cup)$
4:     $\mathscr{D} = (\mathcal{V}_\Delta, \mathbf{F}_\Delta)$
5:     **return** $\mathscr{D}$
6: **end function**

    **Equation 3 Implementation:**
7: $\mathscr{D}_{\text{delta}} \leftarrow (\emptyset, \mathbf{0})$                                    $\triangleright$ Initialize with empty set
8: **for** $i \leftarrow 1$ **to** $N$ **do**                              $\triangleright$ $N$: #Input frame
9:     // $\ominus$: frame differencing
10:     $\mathscr{D}_{\text{tmp}} \leftarrow$ SPARSEDELTA$(\mathscr{D}_t, \mathscr{D}_{t-i}, \ominus)$
11:     // $\oplus$: motion cue fusion
12:     $\mathscr{D}_{\text{delta}} \leftarrow$ SPARSEDELTA$(\mathscr{D}_{\text{delta}}, \lambda^{i-1}\mathscr{D}_{\text{tmp}}, \oplus)$
13: **end for**

---

**Backbone-Decoder Network** We implement the 3D backbone MinkowskiNet18 network architecture in Backbone($\cdot$), using the $\mathtt{spconv}$ library[2], following the design in Fig. 4 of [9]. The detailed implementation of Decoder($\cdot$) is described below, following [44].

$$\mathbf{Z}_i = \sigma\left(\text{Conv}_{1d}\left([\mathbf{H}_{i-1}, \mathbf{F}_{t-1}], \mathbf{W}_z\right)\right)$$

$$\mathbf{R}_i = \sigma\left(\text{Conv}_{1d}\left([\mathbf{H}_{i-1}, \mathbf{F}_{t-1}], \mathbf{W}_r\right)\right)$$

$$\tilde{\mathbf{H}}_i = \tanh\left(\text{Conv}_{1d}\left([\mathbf{R}_i \odot \mathbf{H}_{i-1}, \mathbf{F}_{t-1}], \mathbf{W}_h\right)\right)$$

$$\mathbf{H}_i = \mathbf{Z}_i \odot \mathbf{H}_{i-1} + (1 - \mathbf{Z}_i) \odot \tilde{\mathbf{H}}_i$$

$$\Delta\hat{\mathcal{F}} = \text{MLP}(\mathbf{H}, \mathbf{F}_{t-1}),$$

where $\mathbf{H}_{i-1}$ represents the previous hidden state at the $i$-th iteration. In the first iteration, $\mathbf{H}_0$ is initialized using V2P($\mathscr{D}_{(\text{out})}$) : $\text{R}^{V \times C} \rightarrow \text{R}^{N_{t-1} \times C}$ as described in the main paper. The tensors $\mathbf{Z}$, $\mathbf{H}$, and $\mathbf{F}$ all have dimensions $\mathbb{R}^{N_{t-1} \times C}$, where $N_{t-1}$ is the number of points and $C$ is the feature dimension. $\mathbf{W}_z, \mathbf{W}_r$, and $\mathbf{W}_h$ are trainable weight parameters in the convolutional gated recurrent unit (GRU). $\mathbf{H}$ is the final output of GRU.

## A.2 Training Settings

**For leaderboard experiments,** Argoverse 2 [39] test set results are directly obtained from the public leaderboard [1] to ensure a fair comparison. In the public leaderboard setting, evaluation is conducted within a $70 \times 70$ m area (or a $35$ m perception range) around the ego vehicle. To align with this, $\Delta$Flow is initially trained on a $76.8 \times 76.8$ m grid, corresponding to a $38.4$ m perception range. The voxel grid size is $512 \times 512 \times 32$ with voxel resolution set to $(0.15, 0.15, 0.15)$ m in our best-performing configuration. The number of input frames is set to $5$, with a time decay factor $\lambda = 0.4$. The model is trained using the Adam optimizer [27], with a batch size of 20 across 10 NVIDIA 3080 GPUs for around 18 hours over 21 epochs. We use a cosine decay learning rate schedule with a linear warmup. The learning rate reaches a target of $2 \times 10^{-3}$ after the 2-epoch warmup phase and then decays to a minimum of $2 \times 10^{-4}$.

**For Waymo and other local experiments,** all baselines are retrained and reproduced under the same device settings to ensure consistent evaluation. To match default settings in prior methods, all models,

---

[2]https://github.com/traveller59/spconv

including ours, are trained with a voxel resolution of $0.2$ m, a spatial range of $51.2$ m, a fixed total of 15 epochs, and the same training augmentation on the same computing cluster. We used a batch size of 32, a fixed learning rate of $4 \times 10^{-3}$, and trained on four NVIDIA A100 GPUs for all models. Runtime evaluations are conducted on a desktop system equipped with an Intel i7-12700KF processor and a single NVIDIA RTX 3090 GPU.

**For all experiments,** to improve robustness against elevation variations and sensor viewpoint changes, we apply random height augmentation with 80% probability (uniform offset $\in [0.5, 2.0]$m along $z$-axis) and random flipping along the $xy$-axis with a 20% probability per iteration during all training.

For loss formulations, we assign category weights $w_c = [1.0, 1.5, 2.0, 2.5]$ corresponding to the meta-categories $c = $ [cars, other vehicles, pedestrians, VRUs] as defined by Argoverse 2 [39]. We also apply speed-dependent weights $\gamma_b = [0.1, 0.4, 0.5]$ for static ($v < 0.4$m/s), slow-moving ($0.4 \leq v < 1.0$m/s), and dynamic ($v \geq 1.0$m/s) objects. These weights are experimentally determined with a focus on safety prioritization. Higher values are assigned to vulnerable road users (VRUs) and pedestrians to reflect their critical safety importance, as well as to dynamic objects. In the future, the weighting scheme within the loss function may be guided by a multi-task learning strategy, such as the one proposed in [32, 42].

# B    Additional Analysis

## B.1    $\Delta$ Scheme Efficiency Analysis

Based on Brent's theorem [4], the parallel time complexity for the sparse $\Delta$ scheme in Algorithm 1 is $\frac{O(|\mathcal{V}_\Delta| \log(|\mathcal{V}_\Delta|))}{N_p} + \log^2(|\mathcal{V}_\Delta|)$, where $|\mathcal{V}_\Delta|$ represents the number of voxels containing points, and $N_p$ denotes the number of parallel threads. In multi-frame settings, while the number of active voxels $|\mathcal{V}_\Delta|$ increases with more frames, its growth rate remains low relative to the dense format, as reflected in the ratio shown in Table 6. Consequently, the time consumption for the sparse $\Delta$ scheme exhibits minimal variation across different frame settings in

Table 6: Comparison of the average number of voxels in $\mathscr{D}_{\texttt{delta}}$ across different frame settings (Sparse vs. Dense Representation) in Argoverse 2 validation set. The storage ratio is calculated as $\frac{\text{\#Active Voxels}}{\text{\#Dense Voxels}} \times 100\%$. The dense baseline assumes a resolution of $X \times Y \times Z = 512 \times 512 \times 32$.

| Frame | #Active Voxels | Storage Ratio |
|-------|----------------|---------------|
| 2     | 29475          | 0.35%         |
| 5     | 45310          | 0.54%         |
| 10    | 63745          | 0.76%         |
| 15    | 78666          | 0.94%         |
| Dense | 8388608        | 100.00%       |

Fig. 3 of the main paper, consistent with our parallel time complexity analysis. In comparison with the dense matrix operation, which requires a parallel time complexity of $\frac{O(|\mathcal{V}_{\text{dense}}|)}{N_p}$, the sparse operation achieves up to a $10\times$ speedup and $100\times$ memory reduction. This stems from the substantial disparity in the number of voxels to be processed, where $|\mathcal{V}_\Delta| \gg |\mathcal{V}_{\text{dense}}|$ as in Table 6.

## B.2    Temporal Decay Analysis

To integrate motion cues across multiple frames, we introduce a decay factor $\lambda$ in the $\Delta$ scheme, which progressively downweights older frames. Table 7 compares in detail scene flow estimation performance on different $\lambda$ values for both 5-frame and 10-frame settings.

As shown in Table 7, incorporating $\lambda$ consistently improves performance compared to the non-decayed setting ($\lambda$=1). Notably, all decay configurations outperform the previous state-of-the-art method, Flow4D, demonstrating the effectiveness of our approach regardless of the specific $\lambda$ choice.

Different $\lambda$ values impact performance in distinct ways. A smaller $\lambda$ (e.g., 0.4) reduces errors for fast-moving objects (e.g., cars), while a larger $\lambda$ (e.g., 0.8) improves static background estimation (Mean Three-way EPE). This suggests that a lower $\lambda$ downweights older frames, benefiting fast motion, whereas a higher $\lambda$ preserves historical context, enhancing stability in static regions. Further analysis on optimal $\lambda$ selection based on scenario dynamics will be explored in future work.

## B.3    Loss Function

Table 8 compares different combinations of different loss items: motion-based $\mathcal{L}_{\text{deflow}}$[44], category-balanced, and instance-consistency, evaluated using Dynamic Bucket-Normalized metrics and three-way EPE.

Table 7: Ablation study of the time decay factor $\lambda$ in $\Delta$Flow, evaluated on the Argoverse 2 validation set with 5 and 10 input frames.

| #f | $\lambda$ / Method | Dynamic Bucket-Normalized ↓ | | | | | Three-way EPE (cm) ↓ | | | |
|---|---|---|---|---|---|---|---|---|---|---|
| | | Mean | CAR | OTHER | PED | WHE | Mean | FD | FS | BS |
| 5 | 0.2 | 0.1905 | 0.1488 | 0.1725 | 0.2163 | 0.2244 | 3.31 | 7.85 | 1.35 | 0.74 |
| | 0.4 | **0.1901** | **0.1479** | 0.1723 | 0.2160 | **0.2243** | 3.31 | 7.85 | 1.35 | 0.74 |
| | 0.6 | 0.1926 | 0.1501 | **0.1670** | 0.2182 | 0.2352 | 3.30 | 7.88 | 1.31 | 0.72 |
| | 0.8 | 0.1915 | 0.1482 | 0.1750 | **0.2137** | 0.2291 | **3.26** | **7.84** | 1.24 | 0.70 |
| | 1 | 0.2024 | 0.1493 | 0.1876 | 0.2268 | 0.2460 | 3.34 | 7.98 | 1.31 | 0.74 |
| | Flow4D [18] | 0.2147 | 0.1631 | 0.1767 | 0.2522 | 0.2667 | 3.59 | 8.49 | 1.39 | 0.89 |
| 10 | 0.2 | 0.1917 | 0.1537 | 0.1846 | 0.2070 | 0.2217 | 3.35 | 8.02 | 1.25 | 0.78 |
| | 0.4 | **0.1901** | **0.1500** | 0.1853 | 0.2010 | **0.2241** | 3.30 | 7.94 | 1.23 | 0.73 |
| | 0.6 | 0.1913 | 0.1514 | 0.1873 | 0.2021 | 0.2245 | 3.36 | 8.06 | 1.25 | 0.76 |
| | 0.8 | 0.1954 | 0.1557 | **0.1789** | **0.2001** | 0.2471 | **3.26** | **7.84** | 1.24 | 0.70 |
| | 1 | 0.1967 | 0.1505 | 0.1847 | 0.2087 | 0.2431 | 3.37 | 8.14 | 1.24 | 0.72 |
| | Flow4D [18] | 0.2022 | 0.1494 | 0.1707 | 0.2284 | 0.2603 | 3.45 | 8.09 | 1.46 | 0.81 |

Table 8: Ablation study of proposed loss items. Results are evaluated on the Argoverse 2 validation set using the $\Delta$Flow model ($\lambda = 0.8$) with 5 input frames. **Bold** indicates the best performance, underline marks the second-best, and red highlights settings with a significant performance drop.

| $\mathcal{L}_{\text{deflow}}$ | Loss item category | instance | Dynamic Bucket-Normalized ↓ | | | | | Three-way EPE (cm) ↓ | | | |
|---|---|---|---|---|---|---|---|---|---|---|---|
| | | | Mean | CAR | OTHER | PED | VRU | Mean | FD | FS | BS |
| ✓ | | | 0.2094 | 0.1504 | 0.1854 | 0.2398 | 0.2618 | 3.32 | 8.04 | 1.25 | 0.66 |
| ✓ | ✓ | | 0.1962 | 0.1511 | 0.1732 | 0.2148 | 0.2457 | 3.27 | 7.94 | **1.22** | **0.65** |
| ✓ | | ✓ | 0.1971 | 0.1501 | 0.1675 | 0.2238 | 0.2471 | 3.38 | 7.94 | 1.45 | 0.75 |
| | ✓ | ✓ | **0.1881** | **0.1482** | **0.1609** | 0.2151 | **0.2280** | 3.93 | **7.84** | 1.27 | 2.69 |
| ✓ | ✓ | ✓ | 0.1915 | **0.1482** | 0.1750 | **0.2137** | 0.2291 | **3.26** | **7.84** | 1.24 | 0.70 |

The baseline model includes only the motion-based loss [44], which already achieves promising results. Adding the category-balancing loss significantly improves performance for underrepresented categories, reducing pedestrian error from 0.240 to 0.215 and VRU error from 0.262 to 0.246. This stems from the increasing weight of these objects in the category-balancing loss. However, increasing the weight of these categories slightly lowers accuracy for larger objects, such as cars. Despite this trade-off, the overall mean bucket-normalized error improves from 0.209 to 0.196.

Incorporating the Instance Consistency Loss improves performance across all dynamic objects, reducing the mean dynamic bucket-normalized error from 0.209 to 0.197 by enforcing consistency across moving instances. However, this comes with a slight increase in three-way EPE, mainly on static points.

When using only the category-balanced and instance consistency losses without motion-based supervision, the mean dynamic bucket-normalized error drops to 0.188, the best among all combinations. However, the three-way EPE increases sharply from 3.26 to 3.93 cm, mainly due to a spike in background static error (from 0.70 to 2.69 cm). This highlights the importance of motion-based supervision in constraining overall scene flow predictions.

The best balance between mean Dynamic Bucket-Normalized and Three-way EPE is achieved when all three items are combined. The mean Dynamic Bucket-Normalized error reaches 0.192, and pedestrian error decreases by approximately 11% compared to the motion-only baseline (0.240 to 0.214). Importantly, the three-way EPE remains stable or slightly improves (3.26 vs. 3.32 cm), showing that our full loss formulation enhances learning for smaller, dynamic objects without compromising motion estimation for the rest of the scene. These results validate the effectiveness of our final loss design in improving accuracy.

## B.4   Cross-domain Generalization

This section provides a detailed per-category breakdown of the cross-domain generalization results discussed in Section 5.4. As shown in Table 9, the evaluation is conducted using the Dynamic Bucket-

Normalized EPE metric. The results confirm that ΔFlow maintains state-of-the-art generalization performance across diverse object categories, such as CAR, PED, and VRU, in both cross-domain settings. Notably, the largest gains are observed in the pedestrian and VRU categories, where motion and sensor-domain differences are the most significant.

Table 9: Detailed cross-domain generalization results using the Dynamic Bucket-Normalized EPE metric (lower is better). Performance is shown for Argoverse 2 (2×32-channel) → Waymo (64-channel) and vice versa. The 'OTHER' vehicle category is not labeled in the Waymo dataset and is therefore excluded from the Waymo evaluation. Our method consistently outperforms competitors across all object categories in both cross-domain scenarios.

| Methods | Dynamic Bucket-Normalized ↓ | | | | | | | | |
|---|---|---|---|---|---|---|---|---|---|
| | Argoverse 2 (2x32) → Waymo (64) | | | | Waymo (64) → Argoverse 2 (2x32) | | | | |
| | Mean | CAR | PED | VRU | Mean | CAR | OTHER | PED | VRU |
| SeFlow | 0.423 | 0.252 | 0.626 | 0.391 | 0.400 | 0.269 | 0.349 | 0.559 | 0.421 |
| NSFP | 0.574 | 0.315 | 0.823 | 0.584 | 0.597 | 0.427 | 0.319 | 0.915 | 0.728 |
| DeFlow | 0.346 | 0.156 | 0.545 | 0.339 | 0.326 | 0.201 | 0.267 | 0.434 | 0.400 |
| Flow4D | 0.217 | **0.091** | 0.424 | 0.135 | 0.205 | 0.158 | 0.195 | 0.218 | 0.246 |
| ΔFlow (Ours) | **0.198** | **0.091** | **0.395** | **0.109** | **0.194** | **0.155** | **0.184** | **0.203** | **0.234** |

## B.5 Qualitative Results

The qualitative results in the main paper are derived from the scenes 'fbd62533-2d32-3c95-8590-7fd81bd68c87' and 'dfc32963-1524-34f4-9f5e-0292f0f223ae' in the Argoverse 2 validation set. Here, we present additional qualitative results, comparing ground truth, the previous state-of-the-art method Flow4D [18], and our approach ΔFlow.

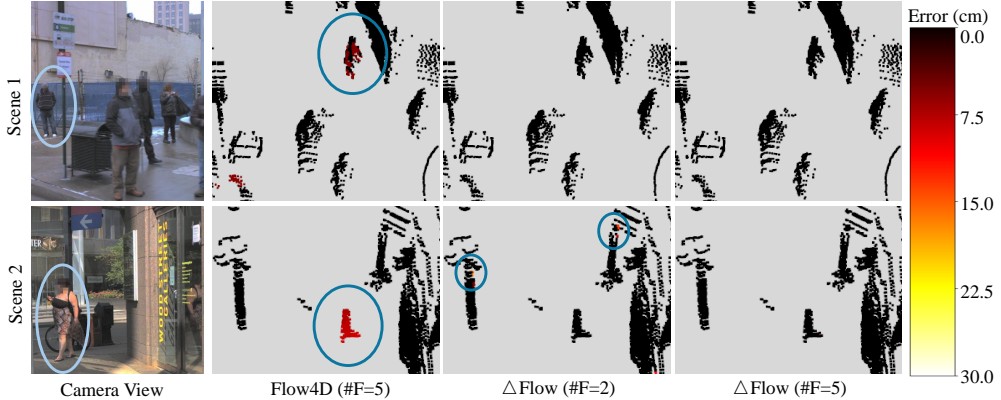

Figure 9: Error maps comparison of 3D flow prediction from the Argoverse 2 validation set, focusing on small objects such as pedestrians. The color bar represents the pointwise L2 norm error in centimeters. ΔFlow with '#F=2' denotes a two-frame input, while all others use five frames. The two scenes correspond to scene IDs 'c8ec7be0-92aa-3222-946e-fbcf398c841e' and '9f871fb4-3b8e-34b3-9161-ed961e71a6da'.

Figure 9 and Figure 10 illustrate error maps for small (pedestrians) and large (trucks) objects, respectively. In Figure 9, the upper row (Scene 1) shows that ΔFlow with two frames improves motion consistency for pedestrians compared to Flow4D, highlighting the effectiveness of the Instance Consistency Loss. The bottom row (Scene 2) further demonstrates improved detection of under-represented small objects, suggesting that Category-Balanced Loss enhances motion estimation for these classes. Figure 10 evaluates large objects. ΔFlow better captures truck motion, demonstrating improved accuracy for large-scale dynamics.

Across all settings, five-frame ΔFlow outperforms the two-frame variant. This suggests that ΔFlow effectively captures global motion cues across multi-frames while maintaining a stable input size.

The color-flow visualization in Figure 11 further supports these findings. Color shifts from the ground-truth color indicate discrepancies in speed and direction. ΔFlow consistently outperforms

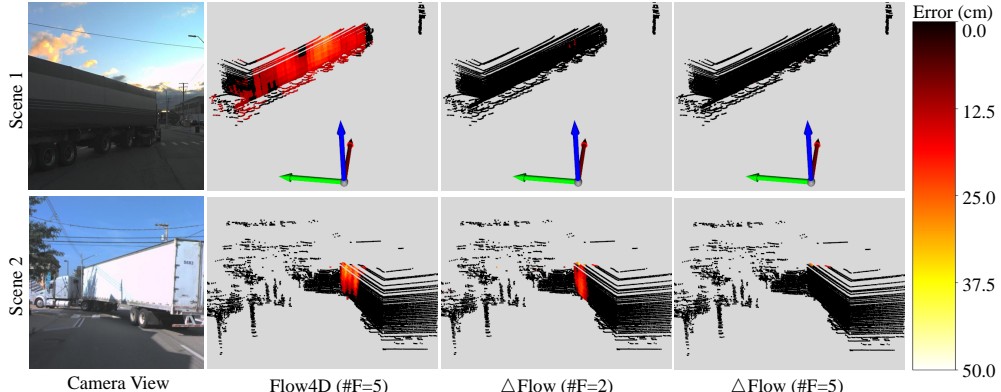

Figure 10: Error maps comparison of 3D flow prediction from the Argoverse 2 validation set, focusing on large vehicles, such as trucks. The color bar represents the pointwise L2 norm error in centimeters. ΔFlow with '#F=2' denotes a two-frame input, while all others use five frames. The two scenes correspond to scene IDs '2c652f9e-8db8-3572-aa49-fae1344a875b' and 'adf9a841-e0db-30ab-b5b3-bf0b61658e1e'.

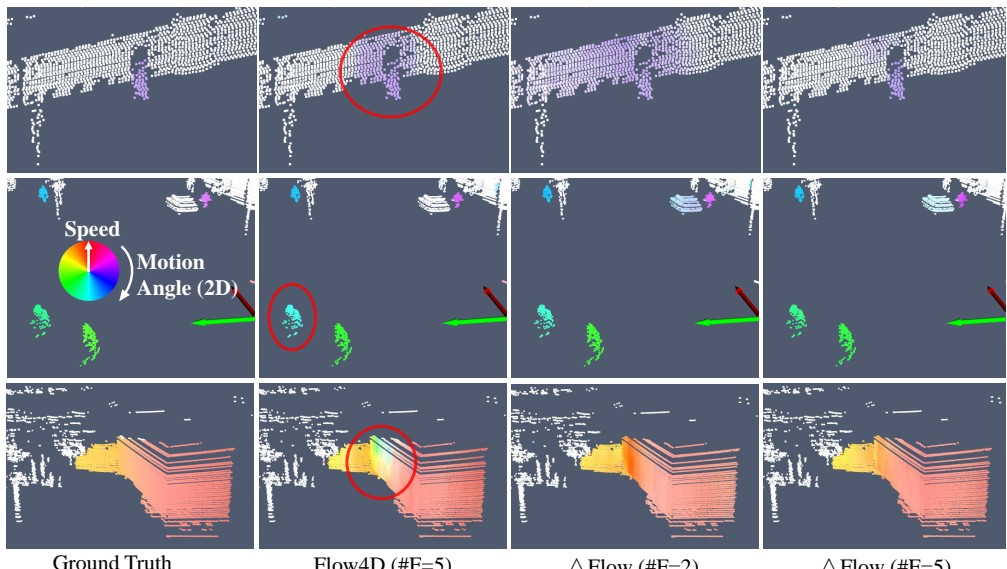

Figure 11: Color-flow visualization comparison of 3D flow prediction from the Argoverse 2 validation set. Direction is encoded as hue, and magnitude as saturation. Ground-truth labels are shown on the left. The three scenes correspond to scene IDs '77574006-881f-3bc8-bbb6-81d79cf02d83', '78f7cb5c-9d51-34f0-b356-9b3d83263c75' and 'adf9a841-e0db-30ab-b5b3-bf0b61658e1e'.

Flow4D across static backgrounds (row 1), small objects (row 2), and large objects (row 3), with five frames yielding the most accurate motion estimates.

## C  Other Discussion

**Broader Impact** Our ΔFlow framework offers scalable and efficient multi-frame 3D motion estimation, enhancing safety and reliability in applications such as autonomous driving, robotic assistance, and augmented reality through improved dynamic scene understanding. Its scalability and computational efficiency reduce overall resource consumption, contributing to more sustainable machine learning system development. However, the same efficient motion tracking capabilities could be misused for pervasive surveillance or unauthorized monitoring, posing privacy risks. Responsible data governance, strict access control, and adherence to ethical deployment standards are essential to ensure the technology is used for societal benefit.

