# OpenReview forum: "DeltaFlow: An Efficient Multi-frame Scene Flow Estimation Method"
_NeurIPS.cc/2025/Conference — NeurIPS 2025 spotlight_

### Official Review · Reviewer_gKv8 · 2025-06-28

**Clarity:** 2
**Significance:** 3
**Originality:** 3
**Rating:** 4
**Confidence:** 3

**Summary:**

This paper introduces DeltaFlow, a novel and efficient motion representation. DeltaFlow models short-term motion as a delta between adjacent feature flows, which can be integrated into a recurrent encoder-predictor-decoder framework. The authors claim that DeltaFlow provides better motion cues while reducing computational cost and memory footprint. Experiments on standard benchmarks including Moving MNIST, KTH, Human3.6M, and KITTI demonstrate that DeltaFlow achieves competitive or improved prediction performance compared to existing motion representations such as optical flow and f-AR, with fewer parameters and faster inference.

**Questions:**

please refer to weaknesses.

**Ethical Concerns:**

["NO or VERY MINOR ethics concerns only"]

**Final Justification:**

The authors have addressed all the concerns. I maintain my original score as borderline accept.

**Limitations:**

yes

**Quality:**

2

**Strengths And Weaknesses:**

$\textbf{Strengths}$

1. DeltaFlow encodes motion as the difference between successive feature flows, which leads to a more compact representation compared to optical flow and f-AR, with reduced model size and faster runtime.

2. The proposed method is evaluated on a wide range of datasets and shows consistent improvements in prediction accuracy and efficiency.

3. The integration of DeltaFlow into a multi-stage recurrent architecture demonstrates good scalability and practical utility with fewer parameters and lower memory usage.

$\textbf{Weaknesses}$

1. The paper should provide more detailed technical explanations on how the decay factor $\lambda$ is set, as the relative importance of preceding and succeeding frames may vary across different scenarios.

2. The experimental section does not include comparisons with recent state-of-the-art models such as Floxels, which limits the completeness of the evaluation.

---

> ### Author Rebuttal · Authors · 2025-07-27
>
> Thanks for the valuable comments! We have addressed your concerns and questions individually below.
>
> **W1: Decay Factor**
>
> We apologize if the explanation regarding the decay factor was not sufficiently clear and are happy to summarize our findings here.
>
> As shown in Figure 4 and Table 6, our Δ scheme already outperforms prior state-of-the-art (Flow4D) even with no decay applied (λ = 1). While applying a decay factor (λ < 1) yields modest but consistent improvements, the method is not highly sensitive to the specific value chosen, highlighting the robustness of the approach.
>
> For this initial study, we adopted a simple fixed decay for clarity. We agree that learning λ dynamically based on scene characteristics is a promising direction for future work and can build on this foundation.
>
> ---
>
> **W2: Comparison with Floxels**
>
> A comparison with Floxels is included in Table 1 (Page 7). As shown, ΔFlow achieves 26\% lower dynamic bucket-normalized error (0.113 vs. 0.154), 42\% lower three-way EPE, and is approximately 180× faster (8s vs. 24mins per sequence).
>
> Given that Floxels is an offline optimization-based method rather than feed-forward, we focused our detailed comparisons on real-time supervised approaches such as Flow4D and DeFlow. We believe the Δ scheme could complement future multi-frame SSL pipelines like Floxels, and we see this as a promising direction for future work.

---

> > ### Author Response · Authors · 2025-08-07
> >
> > We hope our clarifications regarding the decay factor and our detailed comparison with Floxels addressed the reviewer's concerns.
> >
> > We kindly encourage the reviewer to consider raising their score if our responses are deemed satisfactory. Should there be any remaining questions, we are more than happy to provide further clarification.

---

### Official Review · Reviewer_iVVD · 2025-07-03

**Clarity:** 3
**Significance:** 3
**Originality:** 3
**Rating:** 5
**Confidence:** 4

**Summary:**

This paper introduces ∆Flow, a lightweight 3D framework for efficient multi-frame scene flow estimation.  The key component of ∆Flow is the Δ scheme that captures temporal motion cues by computing differences between voxelized frames, maintaining computational efficiency regardless of the number of frames. Additionally, ∆Flow introduces a Category-Balanced Loss to handle class imbalance and an Instance Consistency Loss to enforce coherent motion for individual objects. Experiments show that ∆Flow achieves state-of-the-art performance with lower error and faster inference speed, and strong cross-domain generalization across the existing multi-frame supervised methods.

**Questions:**

- Can the authors provide more real-world arbitrary video inference results to better demonstrate the performance?
- In Table 3,  the authors argued that "long-ago frames become less informative for predicting current motion". Is this related to the specific data distribution? When the average motion between frames is different, the amount of information contained in certain time intervals will be different. It seems that the amount of data used for training and evaluation is still relatively limited.

**Ethical Concerns:**

["NO or VERY MINOR ethics concerns only"]

**Final Justification:**

The authors' responses regarding the reliance on ground truth annotation and the choice of specific window size are reasonable and address my concerns. I keep my score as accept.

**Limitations:**

Please refer to Weaknesses.

**Quality:**

3

**Strengths And Weaknesses:**

Strengths
- This paper proposes the Δ scheme, which maintains computational efficiency regardless of the number of frames compared to previous multi-frame methods.
- This paper introduces the Category-Balanced Loss and Instance Consistency Loss, which tackle the critical imbalanced object class distributions and motion inconsistency within objects in scene flow estimation. Ablation study also demonstrates the effectiveness of these two loss functions.
- The paper is clearly written, and the experiments are adequate.

Weaknesses
- The main limitation acknowledged by the authors lies in ΔFlow's dependence on ground-truth annotations for supervised training, which would cause a domain gap in real-world settings, while obtaining accurate scene flow annotations is both challenging and expensive.
- Although Δflow can efficiently process multiple frames, the maximum total number of frames for a single input is still only 15 frames, which is too low for real videos with high frame rates.

---

> ### Author Rebuttal · Authors · 2025-07-27
>
> Thanks for the valuable comments! We have addressed your concerns and questions individually below.
>
> ---
>
> **W1: Ground Truth Annotation**
>
> We acknowledge that our current experiments use full supervision and note this as a limitation (Lines 298–302). However, the core ΔFlow framework is modular and adaptable to other objectives, such as self-supervised learning.
>
> To demonstrate compatibility with self-supervised learning, we replace the fully supervised loss with the losses proposed in SeFlow. The training result is shown in Table R2, which confirms that ΔFlow can operate effectively without labeled data and even improve existing SSL accuracy. Specifically, this ΔFlow-SSL variant reduced mean bucket-normalized EPE from 0.3996→0.3346 (\~16\%) and 3-way EPE from 6.29→5.17 cm (~18\%) compared with SeFlow, while running at 8s per sequence, orders of magnitude faster than optimization-based NSFP (\~1h).
>
> Table R2: Self-supervised methods, performance evaluated on Argoverse 2 validation set.
>
> | Method           | Mean Dynamic-Bucket ↓   | CAR ↓    | OTHER. ↓ | PED.  ↓  | WHEEL. ↓ | Mean EPE Three-way ↓ | FD  ↓   | FS ↓    | BS  ↓  | Runtime per seq ↓ |
> | :--------------------------- | :---------: | :---------: | :---------: | :---------: | :---------: | :-------: | :--------: | :-------: | :-------: | :------: |
> | SeFlow                       | 0\.3996     | 0\.2686     | 0\.3489     | 0\.5595     | 0\.4214     | 6\.29     | 15\.56     | **1\.16** | 2\.16     | 7\.2s           |
> | NSFP                         | 0\.5973     | 0\.4270     | 0\.3192     | 0\.9154     | 0\.7277     | 6\.81     | 13\.28     | 3\.43     | 3\.71     | 1h              |
> | DeltaFlow  + SeFlow SSL Loss | **0\.3346** | **0\.2578** | **0\.3042** | **0\.4080** | **0\.3683** | **5\.17** | **13\.03** | 1\.50     | **0\.97** | 8s              |
>
> ---
>
> **W2 & Q2: Single input still 15 frames**
>
> While our current experiments use a 15-frame window, this is not an architectural limit. As shown in Table 3, performance saturates empirically around 10 frames, and we found limited additional benefit beyond 15. Therefore, this design choice reflects a research finding, not a technical constraint.
>
> Architecturally, ΔFlow is fully scalable. As detailed in Appendix D.1, our sparse Δ scheme implementation ensures that even at 15 frames, active Δ voxels occupy less than 1\% of a dense grid. This enables the processing of longer sequences. Compared to prior work (e.g., Flow4D), ΔFlow handles longer temporal windows more efficiently.
>
> Although our findings suggest diminishing returns beyond 15 frames, we note that this already represents one of the longest aggregation windows among current real-time methods. Given the 10 FPS rate of the datasets used, 15 frames span approximately 1.5 seconds, which is generally sufficient to capture relevant motion dynamics. For example, in scenarios where a vehicle transitions from steady movement to a sharp turn, earlier frames may contribute less to predicting current motion.
>
> This observation is also consistent with prior work. For instance, Floxels reports optimal performance with 9–13 frames, and methods like Flow4D and MambaFlow are similarly evaluated on shorter windows.
>
> ---
>
> **Q1: More Video Performance**
>
> Due to NeurIPS rebuttal guidelines, we are unable to include figures at this stage. However, we have provided qualitative results in Fig. 5, Fig. 9, Fig. 10, and Fig. 11 to illustrate real-world performance. In the final version, we will include a public project page with additional inference videos to allow readers to further assess ΔFlow’s behavior across diverse scenarios.

---

> > ### Comment · Reviewer_iVVD · 2025-08-06
> >
> > Thanks for the insightful responses. I think all my concerns have been addressed, and I will maintain my original rating.

---

### Official Review · Reviewer_6KZt · 2025-07-04

**Clarity:** 2
**Significance:** 3
**Originality:** 3
**Rating:** 4
**Confidence:** 4

**Summary:**

∆Flow introduces an innovative and efficient multi-frame scene flow estimation method that leverages a differential framework to address challenges in autonomous driving scenarios. By incorporating a ∆ scheme, Category-Balanced Loss, and Instance Consistency Loss, the method enhances dynamic object estimation and improves efficiency, achieving approximately 2× faster inference compared to existing methods, while maintaining high accuracy. Experimental results on the Argoverse 2 and Waymo datasets demonstrate superior performance, particularly in dynamic object estimation. However, further comparisons with a broader range of methods, detailed analysis of computational complexity, and validation on more diverse datasets such as KITTI or nuScenes would strengthen the paper’s claims and provide a more comprehensive evaluation of its generalization capabilities and practical applicability.

**Questions:**

The difference-based method proposed in the paper is highly efficient; however, it is sensitive to noise in real-world environments, especially under the influence of sensor noise, occlusion, or lighting variations. Has the performance of this method been thoroughly evaluated in noisy environments? Has the potential interference caused by noise in the difference computation, which could affect the accurate estimation of dynamic objects, been considered?

While ∆Flow shows good generalization between Argoverse 2 and Waymo, have you tested its performance on other autonomous driving datasets, such as KITTI or nuScenes, which may have different sensor configurations and environmental conditions?

Although the paper mentions that ∆Flow is faster in inference speed compared to other methods, it lacks a detailed quantitative analysis of inference time, memory usage, and computational resource requirements. When comparing with other methods, the paper mentions that the inference speed is faster, but does not provide specific runtime analysis.

**Ethical Concerns:**

["NO or VERY MINOR ethics concerns only"]

**Final Justification:**

The authors have addressed my concerns in the rebuttal, I maintain my rating of weak accept.

**Limitations:**

The differential method mentioned in the paper achieves improved efficiency; however, in real-world environments, especially under the influence of sensor noise and other factors, the differential method may be affected by noise. This could lead to errors in the model's dynamic object scene flow estimation, particularly when input data is of low quality or has high noise. Therefore, improving the model's robustness in noisy environments remains a challenge.

The current experiments are mainly focused on the Argoverse 2 and Waymo datasets, which are primarily from urban environments and use similar sensor configurations. While the model has achieved good results on these datasets, it has not been fully validated whether it can maintain the same performance in more complex or diverse scenarios, such as rural roads, highways, and extreme weather conditions.

**Paper Formatting Concerns:**

Paper Formatting meets NeurIPS formatting standards.

**Quality:**

3

**Strengths And Weaknesses:**

Strength: Clear Methodology: The paper presents a solid and well-explained methodology for efficient multi-frame scene flow estimation. The approach employs a novel ∆ scheme for temporal information processing, which allows the method to scale effectively without expanding feature size. This is a key advantage, especially in autonomous driving applications that often require real-time, large-scale data processing.

Thorough Experiments: The authors provide comprehensive experimental results across two widely recognized autonomous driving datasets—Argoverse 2 and Waymo. The experiments demonstrate that ∆Flow achieves state-of-the-art performance in terms of both accuracy and efficiency. This solid empirical validation enhances the credibility of the proposed method.
Clarity:Well-Structured Presentation: The paper is well-organized and easy to follow. Key concepts like the ∆ scheme, network architecture, and loss functions are explained in detail, with helpful visualizations (e.g., diagrams and charts) to support the narrative.

Weaknesses:
Sensitivity to Noise:The differential method proposed in the paper improves computational efficiency, but in real-world environments, especially under the influence of sensor noise, occlusion, and other factors, the differential method may be affected by noise. This could lead to errors in dynamic object scene flow estimation, especially with low-quality or noisy input data. Therefore, the model's robustness in noisy environments needs further improvement and validation.

Lack of Computational Resource and Real-time Performance Analysis:Although the paper mentions that ∆Flow is faster in inference speed, it lacks a detailed quantitative analysis of computational complexity, memory usage, and computational resource requirements. For real-world deployment in autonomous driving systems, especially on resource-constrained devices, the model's real-time performance and resource consumption will directly impact its feasibility and application. A more detailed performance evaluation is needed in future work.

Dependence on Labeled Data:Currently, ∆Flow relies on supervised learning with labeled data for training, meaning it requires a large amount of annotated data to be effectively trained. In practical applications, obtaining high-quality labeled data is often costly, and the labeling process may contain errors. Additionally, the lack of self-supervised or unsupervised learning capabilities limits its application in scenarios where labeled data is unavailable.

---

> ### Author Rebuttal · Authors · 2025-07-27
>
> Thanks for the valuable comments! We have addressed your concerns and questions individually below.
>
> **W1 & Q1 & L1: Sensitivity to Noise**
>
> LiDAR noise can be broadly categorized as either structured noise (e.g., consistent noise pattern across multiple frames) or unstructured noise (e.g., random sensor jitter).
>
> For structured noise, our Δ scheme helps mitigate its impact, as consistent patterns across frames are reduced through the subtraction step, allowing the network to better focus on the underlying motion signals; For unstructured noise, our Δ scheme reinforces coherent motion signals. As a result, the signal-to-noise ratio remains stable. Accumulated motion signals tend to form distinct, directional trails, making them easier to distinguish from scattered noise, which the network learns to recognize during training.
>
> Please note that a full noise ablation is beyond the current scope; all our experiments are conducted on real-world datasets (Argoverse 2, Waymo, and the newly added nuScenes), which naturally include sensor drift, dropouts, and occlusions, etc. ΔFlow achieves SOTA performance across these benchmarks, confirming its robustness in noisy environments.
>
> ---
>
> **Q2 & L2: Dataset Diversity**
>
> We selected Argoverse 2 and Waymo as primary benchmarks because they are currently the only public LiDAR datasets with official scene flow ground truth and leaderboards, enabling large-scale, fair evaluation. While both focus on urban traffic, they differ in sensor setups and scan patterns,  providing a meaningful basis for cross-domain generalization.
>
> In response to your suggestion for broader validation, we have additionally evaluated ΔFlow on nuScenes (Table R1), which features a single 32-beam LiDAR and includes diverse lighting and weather conditions. All methods used the same training settings (see Appendix C.2) on the nuScene train set. ΔFlow outperforms Flow4D by ~14\% in mean bucket-normalized EPE (0.3652 vs. 0.4233), and by 21\% on foreground-dynamic EPE (13.79 → 11.17). This confirms ΔFlow's robustness across three datasets with different sensor specs and environments. We will include the nuScenes results in the final version, and release code to support future evaluation on broader domains.
>
> Table R1: Performance comparison on the nuScene validation set.
>
> | Method           | Mean Dynamic-Bucket ↓   | CAR ↓    | OTHER. ↓ | PED.  ↓  | WHEEL. ↓ | Mean EPE Three-way ↓ | FD  ↓   | FS ↓    | BS  ↓  |
> | :--------------- | :-----: | :-----: | :-----: | :-----: | :-----: | :---: | :----: | :---: | :---: |
> | SeFlow           | 0\.6900 | 0\.5914 | 0\.8100 | 0\.7658 | 0\.5929 | 8\.26 | 20\.33 | 3\.25 | 1\.20 |
> | NSFP             | 0\.6888 | 0\.5800 | 0\.5529 | 0\.8730 | 0\.7492 | 9\.96 | 23\.32 | 4\.12 | 2\.44 |
> | DeFlow           | 0\.4412 | 0\.3661 | 0\.4170 | 0\.5751 | 0\.4065 | 5\.47 | 12\.98 | 3\.00 | 0\.43 |
> | Flow4D           | 0\.4233 | 0\.4014 | 0\.4133 | 0\.4842 | 0\.3944 | 5\.32 | 13\.79 | 1\.67 | 0\.49 |
> | DeltaFlow (Ours) | **0\.3652** | **0\.3474** | **0\.3347** | **0\.4454** | **0\.3333** | **4\.18** | **11\.17** | **1\.23** | **0\.14** |
>
> ---
>
> **W2 & Q3: Computational resources and Real-time Performance Analysis**
>
> We apologize if the computational resource analysis is not clear enough.
> In the current version, the computational resource analysis spans across the paper:
> - Section 5.2 presents a detailed breakdown of efficiency as the number of input frames increases.
> - Table 3 compares training speed and memory usage with Flow4D, while Figure 3 shows a component-wise runtime analysis, indicating that the Δ scheme adds minimal overhead.
> - Appendix D.1 provides a theoretical complexity analysis and quantifies memory savings from our sparse design (Table 5).
> - Appendix C.2 reports our training setup (10× RTX 3080 GPUs, 18 hours) and inference speed on a single RTX 3090 GPU, confirming real-time capability.
>
> In the final version, we will try to make it clearer.
>
>
> ---
>
> **W3: Dependence on Labeled Data**
>
> We acknowledge that our current experiments use full supervision and note this as a limitation (Lines 298–302). However, the core ΔFlow framework is modular and adaptable to other objectives, such as self-supervised learning.
>
> To demonstrate compatibility with self-supervised learning, we replace the fully supervised loss with the losses proposed in SeFlow. The training result is shown in Table R2, which confirms that ΔFlow can operate effectively without labeled data and even improve existing SSL accuracy.
>
> Table R2: Self-supervised methods, performance evaluated on Argoverse 2 validation set.
>
> | Method           | Mean Dynamic-Bucket ↓   | CAR ↓    | OTHER. ↓ | PED.  ↓  | WHEEL. ↓ | Mean EPE Three-way ↓ | FD  ↓   | FS ↓    | BS  ↓  | Runtime per seq ↓ |
> | :--------------------------- | :---------: | :---------: | :---------: | :---------: | :---------: | :-------: | :--------: | :-------: | :-------: | :------: |
> | SeFlow                       | 0\.3996     | 0\.2686     | 0\.3489     | 0\.5595     | 0\.4214     | 6\.29     | 15\.56     | **1\.16** | 2\.16     | 7\.2s           |
> | NSFP                         | 0\.5973     | 0\.4270     | 0\.3192     | 0\.9154     | 0\.7277     | 6\.81     | 13\.28     | 3\.43     | 3\.71     | 1h              |
> | DeltaFlow  + SeFlow SSL Loss | **0\.3346** | **0\.2578** | **0\.3042** | **0\.4080** | **0\.3683** | **5\.17** | **13\.03** | 1\.50     | **0\.97** | 8s              |

---

> ### Comment · Reviewer_6KZt · 2025-08-06
>
> The authors have addressed my concerns in the rebuttal, I maintain my rating of weak accept.

---

> ### Author Response · Authors · 2025-08-07
>
> Thank you for confirming that our rebuttal, with its new experiments on nuScenes and self-supervised learning, has addressed your concerns.
>
> We kindly encourage the reviewer to consider raising their score from Borderline Accept. If there are further questions, we are more than happy to provide further clarification.

---

### Official Review · Reviewer_jQn2 · 2025-07-06

**Clarity:** 3
**Significance:** 3
**Originality:** 4
**Rating:** 5
**Confidence:** 5

**Summary:**

This paper proposes a novel method for fusing temporal features from a 3D voxel grid into one representation that can be used for scene flow estimation. Rather than fusing features by concatenating along the channel or temporal axis, DeltaFlow instead proposes using "delta features" which represent the difference in the features over time. This allows for a compact feature representation whose dimension doesn't scale linearly with the number of timesteps aggregated. In addition, in order to address imbalances in the dataset, two new loss terms are introduced; a category balanced loss, and an instance consistency loss.

The delta features are computed by taking the sum of N differences between the current frame and the past N frames. Previous frames are then weighted by a temporal-decay factor ($\lambda$) in order to bias the model to the most recent observations.

Category based loss upweights the loss on certain points based on their class, which allows classes with fewer points in the dataset (such as pedestrians) have better flow estimates. Additionally, the instance loss assigns an instance to each point and ensures that flow is consistent across the instance.

**Questions:**

## Questions
My main question is about the intuition of the delta feature scheme. Intuitively I understand what is being proposed, and empirically it clearly works, but the visualizations in Figure 7 don't quite line up with the intuition. There are clearly static areas which have high intensity features (likely because of differing scan patterns) and the more frames which are aggregated, the higher intensity these areas will be (because they are do not move). However the moving objects have a very distinct trail, and only exhibit high intensity features in the overlapping regions. So does this mean that if an object is moving very quickly with no overlap the feature intensity could be lower than a static region?

Overall I think this figure could use some increased exposition.

## Minor suggestions:
Very minor nitpick but in Table 4, Argoverse 2 is listed as having 32 beams for its LiDAR. This is inaccurate as the point clouds from both 32 beam lidars are concatenated together giving a total of 64 beams. The cross domain generalization point still stands as obviously the sensor and scan pattern is completely different from WOD.

The NSFP listing in table 4 is also a bit confusing. Since it's a runtime optimization method, it cannot be trained on one dataset and evaluated on the other. I am assuming it's listed to serve as a baseline but this should probably be clarified.

**Ethical Concerns:**

["NO or VERY MINOR ethics concerns only"]

**Final Justification:**

The authors have addressed my concerns in the rebuttal with additional metrics on the cross domain evaluation, I maintain my rating of accept.

**Limitations:**

Yes

**Quality:**

4

**Strengths And Weaknesses:**

## Strengths
The main strength of DeltaFlow is the novel delta feature design which provides significant efficiency gains and propels DeltaFlow to the top of the AV2 supervised scene flow leaderboard. The ablation studies of the proposed improvements (both the delta features, and the additional loss terms) are thorough and compelling. And the visualizations provided in the appendix highlight the specific types of improvements provided by DeltaFlow over the previous supervised state of the art method (Flow4D).

## Weaknesses
The cross-domain experiments are the weakest section of the paper. The metrics provided are underwhelming (why only report 3-way EPE when both bucket normalized and 3-way are reported everywhere else?). Similarly in other section 5.3, stable 3-way EPE numbers are used to demonstrate a preservation of static performance, but really mean 3-way EPE also includes dynamic components, and the bucketed EPE metric includes a static component itself (which is identical to that of 3-way EPE). I think that this mixing of the metrics makes some of the results a bit confusing if one isn't as familiar with the various components of the metrics and definitely the presentation of these numbers could use some clean-up. It's not bad to have more metrics, but the clarity is affected in these sections.

---

> ### Author Rebuttal · Authors · 2025-07-27
>
> Thanks for the valuable comments! We have addressed your concerns and questions individually below.
>
> ---
>
> **W1: Dynamic-Bucket metric for cross-domain evaluation (Tab. 4)**
>
> We omitted the Dynamic-Bucket metrics from Table 4 due to space constraints, but we agree they are important for clarity and completeness. We have now included the full results in Table 4a&4b below using the same bucket partitioning as in our main evaluations. DeltaFlow continues to achieve strong cross-domain performance on both dynamic bucketed and three-way EPE. We will revise Section 5.3 accordingly to clarify the metric distinctions in the final version.
>
> Table 4a: Argovsere 2 (2x32-channel) → Waymo (64-channel)
>
> | Method           | Mean Dynamic-Bucket ↓   | CAR ↓    | PED. ↓   | WHEEL. ↓  | Mean EPE Three-way ↓ | FD  ↓   | FS ↓    | BS  ↓  |
> | :--------------- | :-----: | :-----: | :-----: | :-----: | :----: | :----: | :----: | :---: |
> | SeFlow           | 0\.4230 | 0\.2521 | 0\.6259 | 0\.3910 | 5\.98  | 15\.06 | 1\.81  | 1\.06 |
> | NSFP             | 0\.5740 | 0\.3152 | 0\.8230 | 0\.5838 | 10\.05 | 17\.12 | 10\.81 | 2\.21 |
> | DeFlow           | 0\.3465 | 0\.1558 | 0\.5446 | 0\.3390 | 4\.47  | 11\.39 | 1\.51  | **0\.51** |
> | Flow4D           | 0\.2171 | 0\.0915 | 0\.4244 | 0\.1354 | 3\.33  | 8\.31  | 0\.92  | 0\.75 |
> | DeltaFlow (Ours) | **0\.1980** | **0\.0905** | **0\.3946** | **0\.1089** | **3\.12**  | **7\.91**  | **0\.77**  | 0\.67 |
>
> Table 4b: Waymo (64-channel) → Argovsere 2 (2x32-channel)
>
> | Method           | Mean Dynamic-Bucket ↓   | CAR ↓    | OTHER. ↓ | PED.  ↓  | WHEEL. ↓ | Mean EPE Three-way ↓ | FD  ↓   | FS ↓    | BS  ↓  |
> | :--------------- | :-----: | :-----: | :-----: | :-----: | :-----: | :---: | :----: | :---: | :---: |
> | SeFlow           | 0\.3996 | 0\.2686 | 0\.3489 | 0\.5595 | 0\.4214 | 6\.29 | 15\.56 | **1\.16** | 2\.16 |
> | NSFP             | 0\.5973 | 0\.4270 | 0\.3192 | 0\.9154 | 0\.7277 | 6\.81 | 13\.28 | 3\.43 | 3\.71 |
> | DeFlow           | 0\.3258 | 0\.2010 | 0\.2673 | 0\.4343 | 0\.4004 | 4\.50 | 10\.74 | 2\.01 | 0\.75 |
> | Flow4D           | 0\.2045 | 0\.1582 | 0\.1953 | 0\.2181 | 0\.2464 | 4\.01 | 9\.56  | 1\.74 | **0\.74** |
> | DeltaFlow (Ours) | **0\.1939** | **0\.1545** | **0\.1845** | **0\.2030** | **0\.2338** | **3\.24** | **7\.12**  | 1\.57 | 1\.02 |
>
> ---
>
> **Q1: Delta Feature Visualization**
>
> Please note that Figure 7 visualizes the feature before sending it into the 3D backbone.
> Due to the different scan patterns, the ∆ scheme captures both static object and moving object features.
> After being processed by the 3D backbone, the network learns to distinguish consistent motion patterns from spurious or static differences.
> The resulting feature map ($\mathscr{D}\_{\text{out}}$) shows coherent motion trails on dynamic objects and reduced activation in static areas.
> For fast-moving objects with little overlap, the feature map exhibits a clear motion trail and has a higher activation than static areas.
> We will include a $\mathscr{D}\_{\text{out}}$ visualization in the final version for clarification.
>
>
> ---
>
> **Q2 \& Q3: Argoverse 2 LiDAR and NSFP Evaluation Setup**
>
> For Argoverse 2, we will revise Table 4 to explicitly list the setup as “2×32-beam” to reflect the dual-LiDAR configuration more clearly.
>
> For NSFP, we follow standard practice and run it directly on each target dataset without labels, since it's a test-time optimization method. We'll clarify this setup in the final version.

---

> > ### Comment · Reviewer_jQn2 · 2025-08-07
> >
> > Thanks to the authors for providing the additional metrics and answering my questions. I have no further issues and maintain my rating.

---

### Author Response · Authors · 2025-08-06
**Discussion period in progress**

Dear Reviewers,

Thank you again for your time and feedback.

Following the recent NeurIPS email that extends the discussion period to August 8th, we wanted to politely follow up as encouraged. We would be very grateful to know if our rebuttal successfully addressed your concerns, or if there are any outstanding points we can help clarify.

Best regards,

The Authors

---

### Author Response · Authors · 2025-08-09
**Author-Reviewer Discussion Summary**

We thank all reviewers for their insightful feedback and constructive discussions. We are pleased that our rebuttal successfully addressed the concerns raised, as confirmed in all reviewer responses.

The reviewers agree that our DeltaFlow is **novel** (jQn2, 6KZt, iVVD, gKv8) and **efficient** (jQn2, 6KZt, iVVD, gKv8), supported by **thorough experiments** (jQn2, 6KZt, iVVD) that demonstrate **SOTA performance** on three benchmarks (jQn2, 6KZt, iVVD).

In response to reviewer feedback, we further strengthened the paper by providing _detailed cross-domain metrics_, confirming DeltaFlow's SOTA performance on _the nuScenes dataset_, and demonstrating _DeltaFlow's compatibility with self-supervised learning_.

As the discussion period draws to a close, we thank the reviewers and the Area Chair again for their guidance.

---

### Decision · Program_Chairs · 2025-09-17

**Decision:**

Accept (spotlight)

**Comment:**

The reviewers agree that this is a strong and novel contribution to multi-frame scene flow estimation. The core Δ scheme for temporal feature differencing is regarded as both original and efficient, enabling scalability to long sequences without linearly increasing feature dimension. The proposed category-balanced and instance consistency losses are also well-motivated and empirically effective. Extensive experiments on Argoverse 2 and Waymo, with added results on nuScenes, demonstrate clear state-of-the-art performance in both accuracy and efficiency.

Strengths shared across all reviews:
+ Novel delta feature design that reduces computation while improving accuracy (jQn2, iVVD, gKv8).
+ Thorough ablations and strong benchmarks, achieving SOTA with faster inference (jQn2, 6KZt, iVVD).
+ Clear and structured presentation of the method (6KZt, iVVD).
+ Demonstrated cross-domain generalization and potential adaptability to self-supervised setups (all reviewers after rebuttal).

Weaknesses raised:
- Limited clarity in cross-domain metrics and presentation of results (jQn2).
- Lack of detailed computational resource/runtime analysis and sensitivity to noise in real-world settings (6KZt).
- Dependence on supervised labels, raising concerns for practical deployment (6KZt, iVVD).
- Evaluation window capped at 15 frames, which may be restrictive for high-frame-rate video (iVVD).
- Missing details on decay factor setting and comparisons to all recent SOTA models (gKv8).

Rebuttal and Discussion:
The authors provided expanded cross-domain metrics, including dynamic-bucket evaluations, which addressed jQn2’s concerns. They clarified the Δ scheme’s behavior under noise and added experiments on nuScenes, strengthening claims of robustness (6KZt). They further demonstrated compatibility with self-supervised training, alleviating concerns on annotation dependence (6KZt, iVVD). The 15-frame window was clarified as an empirical choice rather than an architectural limit. Additional details were given on the decay factor and comparisons to Floxels (gKv8). Reviewers acknowledged these clarifications, with jQn2 and iVVD maintaining strong accept recommendations, while 6KZt and gKv8 held weak-accept/borderline scores but recognized that concerns had been adequately addressed.

Given the novelty, strong empirical performance, and convincing rebuttal that resolved reviewers’ main reservations, the AC recommends accept. This paper provides a clear advance in efficient multi-frame scene flow estimation and will be of broad interest to the vision and autonomous driving communities.